# A ratchet-like apical constriction drives cell ingression during the mouse gastrulation EMT

Alexandre Francou, Kathryn V Anderson†, Anna-Katerina Hadjantonakis*

Developmental Biology Program, Sloan Kettering Institute, Memorial Sloan Kettering Cancer Center, New York, United States

**Abstract** Epithelial-to-mesenchymal transition (EMT) is a fundamental process whereby epithelial cells acquire mesenchymal phenotypes and the ability to migrate. EMT is the hallmark of gastrulation, an evolutionarily conserved developmental process. In mammals, epiblast cells ingress at the primitive streak to form mesoderm. Cells ingress and exit the epiblast epithelial layer and the associated EMT is dynamically regulated and involves a stereotypical sequence of cell behaviors. 3D time-lapse imaging of gastrulating mouse embryos combined with cell and tissue scale data analyses revealed the asynchronous ingression of epiblast cells at the primitive streak. Ingressing cells constrict their apical surfaces in a pulsed ratchet-like fashion through asynchronous shrinkage of apical junctions. A quantitative analysis of the distribution of apical proteins revealed the anisotropic and reciprocal enrichment of members of the actomyosin network and Crumbs2 complexes, potential regulators of asynchronous shrinkage of cell junctions. Loss of function analyses demonstrated a requirement for Crumbs2 in myosin II localization and activity at apical junctions, and as a candidate regulator of actomyosin anisotropy.

**\*For correspondence:**
hadj@mskcc.org

†Deceased

**Competing interest:** The authors declare that no competing interests exist.

## Editor's evaluation

This study employs live imaging to investigate the movement of mesodermal cells in early mouse embryos. By examining the dynamics of cell behavior in normal and mutant embryos, the authors propose that apical constriction of cells results from pulsed contraction guided by crumbs2 signals. The paper presents beautiful images and adds to the molecular understanding of cell migration during early development.

## Introduction

Epithelial-to-mesenchymal transitions (EMTs) are tissue-level morphogenetic programs necessary for normal embryonic development, cancer progression, and metastasis (*Francou and Anderson, 2020*; *Thiery, 2002*; *Thiery et al., 2009*; *Ye and Weinberg, 2015*). EMT that occurs during embryo gastrulation is an evolutionarily conserved event occurring in amniotes as they generate three definitive germ layers (*Lim and Thiery, 2012*; *Nakaya and Sheng, 2008*; *Thiery et al., 2009*). The onset of gastrulation involves formation of the primitive streak at the posterior of the pluripotent epiblast, at around embryonic (E) 6.5 day in the mouse (*Ferrer-Vaquer et al., 2010*; *Nakaya and Sheng, 2008*; *Williams et al., 2012*; *Figure 1A*). Primitive streak initiation is regulated by the convergence of WNT, BMP, and Nodal signals, which together with FGF signal trigger the EMT process (*Ferrer-Vaquer et al., 2010*; *Morgani and Hadjantonakis, 2020*; *Ramkumar and Anderson, 2011*). The basement membrane present basally adjacent to epiblast cells is broken down at the primitive streak, thereby facilitating the movement of cells out of the epiblast tissue layer, accompanied by changes in their cell

shape, including apical constriction and the basal translocation of cell bodies (*Ramkumar et al., 2016*; *Williams et al., 2012*). Basal cell ingression and migration away from the epiblast epithelium is facilitated through the dissolution of apical junctions. As cells at the primitive streak undergo EMT, more laterally positioned epiblast cells converge toward the midline (*Williams et al., 2012*), replenishing the pool for continued ingression and ensuring epithelial integrity.

Apical constriction is an epithelial cell shape change associated with morphogenetic processes such as tissue bending, cell delamination, and internalization (*An et al., 2017*; *Chung et al., 2017*; *Lecuit and Lenne, 2007*; *Martin and Goldstein, 2014*; *Nishimura et al., 2012*; *Simões et al., 2017*). The dynamics of apical constriction at gastrulation have been studied in invertebrates, particularly in *Drosophila*, and shown to be controlled by apical actomyosin contractility (*Marston et al., 2016*; *Martin and Goldstein, 2014*; *Martin et al., 2009*; *Mason et al., 2013*; *Roh-Johnson et al., 2012*). Apical constriction has been associated with cell ingression during the gastrulation EMT in chick and mouse embryos (*Rozbicki et al., 2015*; *Serrano Nájera and Weijer, 2020*; *Williams et al., 2012*), but how cells dynamically constrict their apical surfaces during EMT, and what regulates this process remain open questions. The process of gastrulation occurs along divergent time-scales (~1 hr vs. >24 hr) in *Drosophila* and mouse, and involves distinct mechanisms and spatial parameters (tissue invagination vs. cell ingression). In the chick embryo, Myosin II plays a role in the formation of the primitive streak, and cell ingression from the epiblast layer (*Chuai et al., 2006*; *Rozbicki et al., 2015*; *Serrano Nájera and Weijer, 2020*; *Voiculescu et al., 2007*; *Voiculescu et al., 2014*). Crumbs2, a protein shown to regulate apical polarity in epithelia, is critical for cell ingression during gastrulation EMT in mouse embryos (*Ramkumar et al., 2016*). Apical Crumbs2 and myosin heavy chain IIB show an anisotropic accumulation in mouse epiblast cells in the vicinity of the primitive streak (*Ramkumar et al., 2016*), reminiscent of patterns controlling apical constriction during *Drosophila* salivary gland development (*Röper, 2012*), suggesting these proteins are potential regulators of apical constriction during mouse EMT. Visualizing the cellular dynamics of the epiblast during mouse gastrulation has been a longstanding challenge due to the internal location of the tissue which limits optical access and visualization at high resolution.

Using live imaging of ex utero cultured mouse embryos, and high-resolution visualization and segmentation of epiblast cell membranes, we performed a dynamic quantitative analysis of the apical constriction associated with ingression during the mouse gastrulation EMT. We observed cells undergoing EMT at the mouse primitive streak in a scattered and apparently stochastic manner. Epiblast cells constricted their apical surfaces in a ratchet-like pulsed fashion through asynchronous shrinkage of their apical cell-cell junctions. By analyzing and quantifying the distribution of apical proteins, we uncovered an anisotropic and reciprocal enrichment of actomyosin network and Crumbs complex proteins. Such reciprocal distribution is consistent with actomyosin and Crumbs complex playing a role in regulating junctional shrinkage and apical constriction. The localization of the apical actomyosin network, as well as two kinases aPKC and Rock1, was perturbed in *Crb2* mutants, thereby identifying them as key components of a putative regulatory network driving apical constriction.

## Results

### Epiblast cells undergo apical constriction and isolated ingression at the mouse primitive streak

The mouse gastrulation EMT occurs at the primitive streak, which, by contrast to the chick, does not form a morphologically distinct domain (*Figure 1A and B*). Columnar epithelial epiblast cells with their apical surfaces facing the inner (amniotic) cavity of the embryo apically constrict and elongate basally as they ingress out of the epiblast tissue layer (*Figure 1A and C*; *Williams et al., 2012*). To visualize the dynamic changes in the shapes of all cells in tissue context at single-cell resolution during the gastrulation EMT, we sought to label the membranes and junctions of all epiblast cells and perform time-lapse imaging. Time-lapse imaging of epiblast cells in the vicinity of the primitive streak of developing mouse embryos is challenging for several reasons. Embryos need to be kept intact and tissues cannot be microdissected as epiblast integrity would be lost. The epiblast is cup-shaped with inherent curvature, and it is the tissue located within the deepest extremity of the embryo (up to 60 μm away from the objective), needing to be imaged through the adjacent visceral endoderm and mesoderm tissue layers (*Figure 1C*). Even light-sheet systems, which can provide in toto visualization of embryos

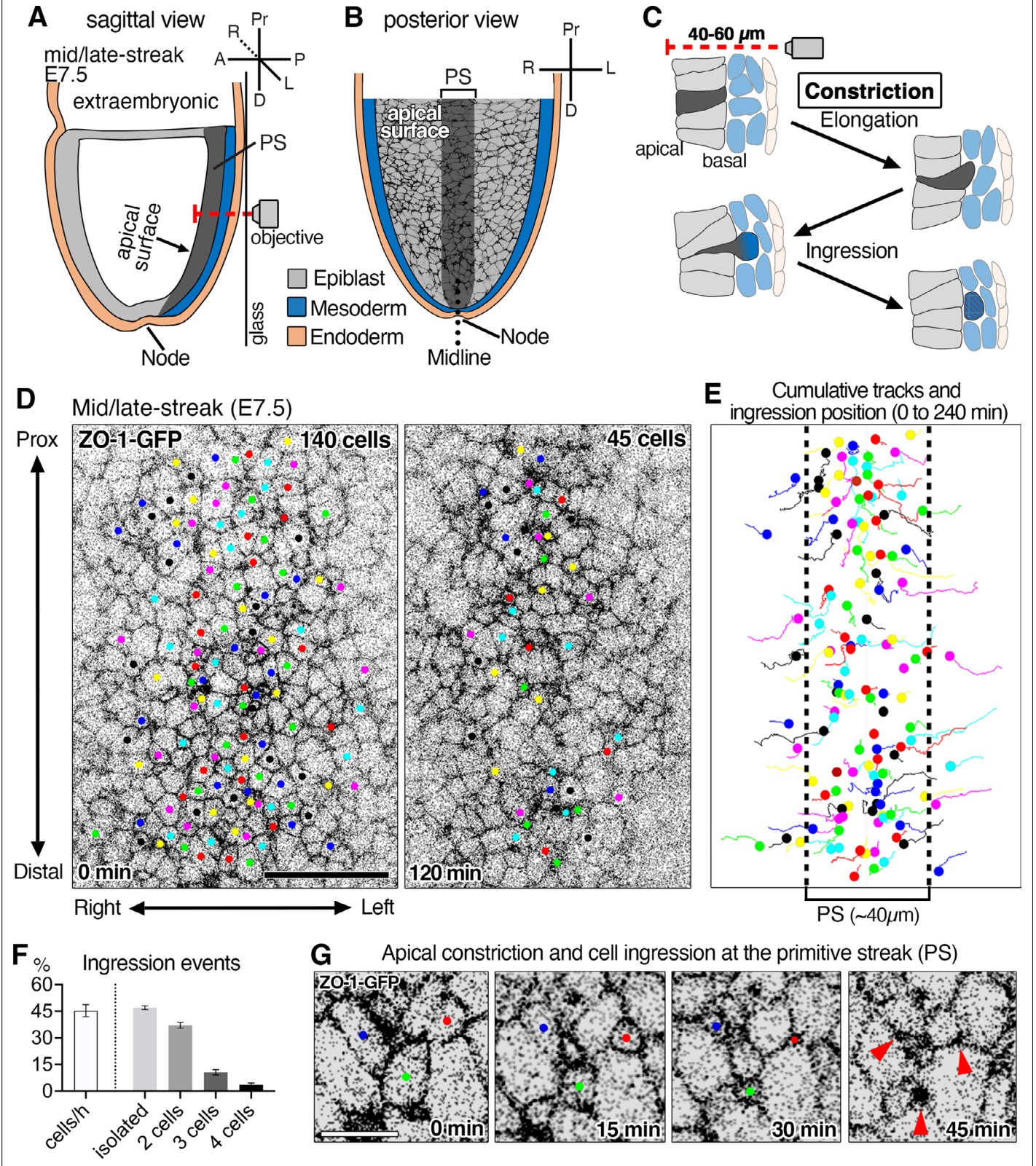

**Figure 1.** Time-lapse imaging of ZO-1-GFP reporter reveals epithelial-to-mesenchymal transition (EMT) events at the primitive streak of the mouse gastrula and apical constriction associated with cell ingression. (**A**) Schematic sagittal section view of embryonic day (E)7.5 mouse embryo and 3D time-lapse imaging performed from the posterior side in glass-bottom dishes with the objective positioned adjacent to the primitive streak (PS). In this configuration, the epiblast apical surface situated furthest from the objective (red dashed line shows the microscope light path). (**B**) Schematic of a view

*Figure 1 continued on next page*

*Figure 1 continued*

(from the inner cavity) of the apical surface of cells within the epiblast layer. The midline (dotted line) separates the right and left sides of the embryo. (**C**) High magnification schematic of a sagittal view of an EMT event at the primitive streak depicting a cell constricting its apical surface, elongating basally (dark gray), and ingressing out of the epiblast layer to integrate into the mesoderm (blue). Note the apical surface of epiblast cells is located 40–60 μm away from the imaging objective. (**D**) Single time points at t=0 min and t=120 min of a time-lapse of a ZO-1-GFP embryo. Movies were analyzed as maximum intensity projections of z-stacks to visualize the apical surfaces of cells. All the cells that can be followed and observed ingressing during 4 hr were tracked. Of 140 cells initially tracked, 95 cells constricted and ingressed over the course of 120 min (45 tracked cells remaining). The time-lapse was acquired with a 5 min time interval, 290 cells were in the field of view, with 140 ingressing cells tracked. (**E**) Cumulative tracking of the 140 cells showing cell tracks over time (lines) and cell position at the time of ingression (dots). Epiblast cells converge toward the primitive streak (~40 μm, dotted lines) where the majority of ingression events occur. (**F**) Graphs showing 44 ± 2.1% of cells in the primitive streak ingress each hour, with 48 ± 1.1% ingressing as single cells, 37 ± 1.7% as pairs of cells, 11 ± 1.5% as triplets (groups of three cells), and 4% as groups of four cells (of a total of 378 ingressing cells analyzed in three embryos). (**G**) High magnification view showing the apical constriction of three cells at the primitive streak. Pr: proximal, D: distal, A: anterior, P: posterior, R: right, L: left, Ext Emb: extra embryonic region, PS: primitive streak. Error bars represent s.e.m. See *Supplementary file 1* for n. Scale bars, D, 40μm; G, 10μm.

The online version of this article includes the following source data and figure supplement(s) for figure 1:

**Source data 1.** Ingression events per hour and percentage of different types of events.

**Figure supplement 1.** Apical constriction and ingression of epiblast cells in the vicinity of the mouse primitive streak.

and have predominantly been used to visualize and segment nuclear-localized reporters (*McDole et al., 2018*), present limitations in imaging membrane and junctional reporters.

We used a ZO-1-GFP protein fusion reporter (*Foote et al., 2013*) to visualize tight junctions, and by extension, the apical surface of epiblast cells. This allowed us to quantify apical surface dynamics (*Figure 1D*). We also used a membrane-localized *Rosa26*^mT/mG reporter to visualize their entire plasma membrane, and to identify completion of the apical constriction associated with cell ingression out of the epithelial epiblast layer (*Figure 1—figure supplement 1A* and *Video 1*). 3D time-lapse imaging of embryos at mid/late-streak stage (E7.5) was performed to image the primitive streak (*Figure 1A and B*). Time-lapse imaging of ZO-1-GFP embryos revealed that epiblast cells undergo extensive rearrangements as cells on the left and right sides of the embryo converge toward the primitive streak (*Figure 1D*, *Figure 1—figure supplement 1C*, *Videos 2 and*

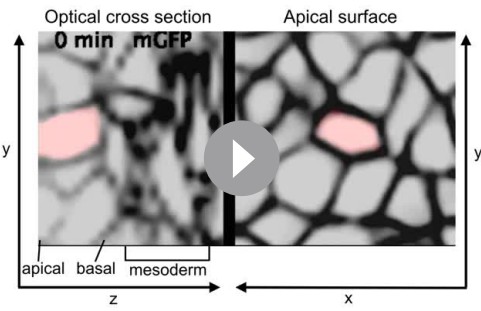

**Video 1.** Time-lapse imaging of a membrane-GFP reporter (recombined *mT/mG*) expressing embryonic day (E)7.5 embryo. Apical constriction (right panel) associated with ingression, as cells move out of the epiblast epithelial layer basally to join the underlying mesoderm (left panel). Images representing single planes of optical sections and apical cell surfaces were denoised and deconvolved. The time-lapse spans 70 min with 10 min time interval between frames.
https://elifesciences.org/articles/84019/figures#video1

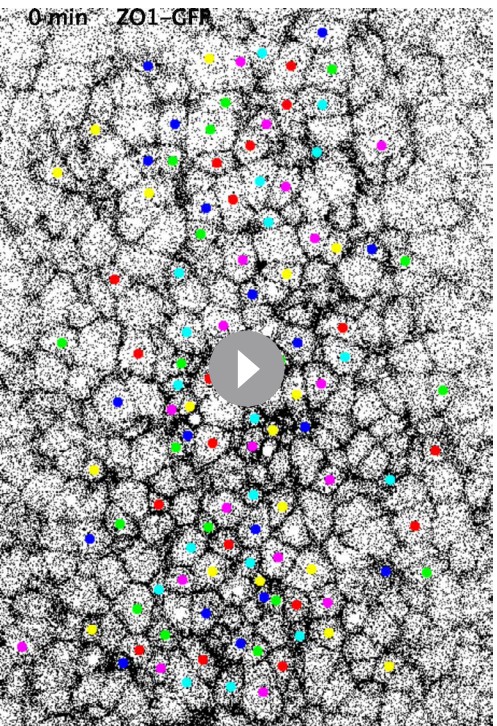

**Video 2.** Time-lapse imaging of the posterior side of a ZO-1-GFP reporter expressing embryonic day (E)7.5 embryo in the vicinity of the primitive streak. Tracking of individual cells reveals lateral epiblast cells converging toward the midline prior to their ingression. Cells around the midline (middle row) do not exhibit extensive movement in the plane of the epithelium. Movie of 240 min with 5 min interval between frames.
https://elifesciences.org/articles/84019/figures#video2

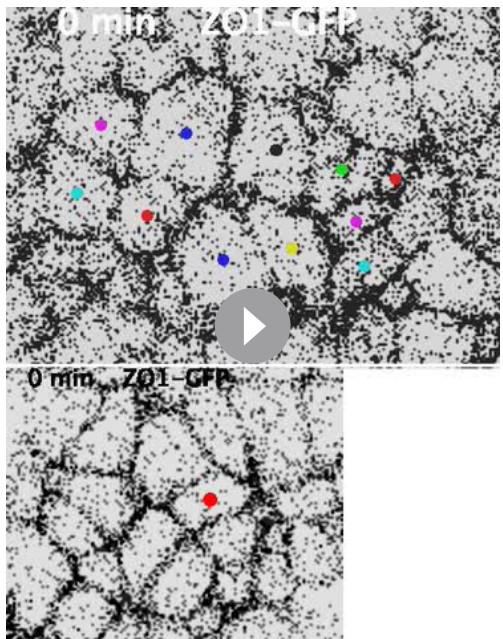

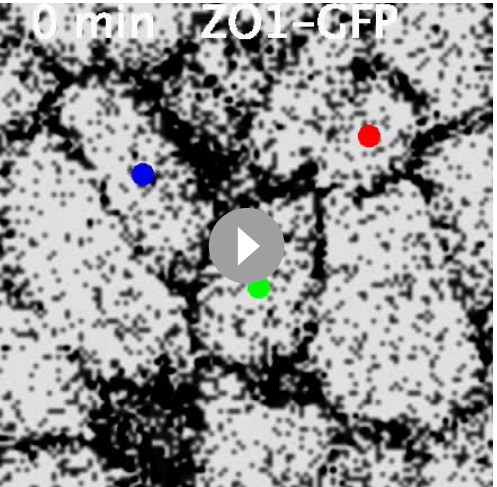

**Video 4.** Time-lapse of ZO-1-GFP highlighting three cells apically constricting and ingressing in the primitive streak region. Sixty min total with 5 min time interval between frames.

https://elifesciences.org/articles/84019/figures#video4

**Video 3.** Time-lapse of ZO-1-GFP reveals dynamic cell behaviors at the primitive streak. Rearrangement of cells in the epiblast near the primitive streak (top panel, 125 min), and cell apical constriction and ingression away from the primitive streak, as the epiblast flows (to the left side) toward the midline (bottom panel, 55 min). Five min time interval between frames.

https://elifesciences.org/articles/84019/figures#video3

*3*; *Williams et al., 2012*). The majority of epiblast cells underwent apical constriction and ingression within an ~40 μm region at the posterior midline (*Figure 1D and E*, *Video 2*). This region corresponded to the domain of Snail expression and basement membrane breakdown within the epiblast (*Francou and Anderson, 2020*; *Ramkumar et al., 2016*; *Williams et al., 2012*), which by convention defines the primitive streak (*Figure 1E*, dotted lines). Rare apical constriction and ingression events were also observed outside the domain of the primitive streak (*Figure 1—figure supplement 1B* and *Video 3*, bottom).

Within a 1 hr period 44 ± 2% of cells within the primitive streak domain, constricted and ingressed (mean ± s.e.m., n=378 cells, three embryos) (*Figure 1D, E and F*, *Video 2*), with ingression events scattered through the streak appearing to occur stochastically. Among the ingressing cells tracked, 48% constricted and ingressed as isolated events (more than 30 min apart from adjacent neighboring cells), while the remaining 52% ingressed as pairs, and occasionally as groups of three to four cells (less than 30 min apart from adjacent neighboring cells) (*Figure 1F*). At the tissue level the epiblast underwent extensive rearrangements, with some cells moving and acquiring new neighbors prior to their coordinate ingression (*Figure 1—figure supplement 1D, E*), while others, which were initially neighbors, ingressed at different times (more than 30 min apart) (*Figure 1—figure supplement 1D, F*).

## Epiblast cells undergo a ratchet-like pulsed apical constriction during ingression

We next focused on changes associated with the apical surface of cells during their ingression.

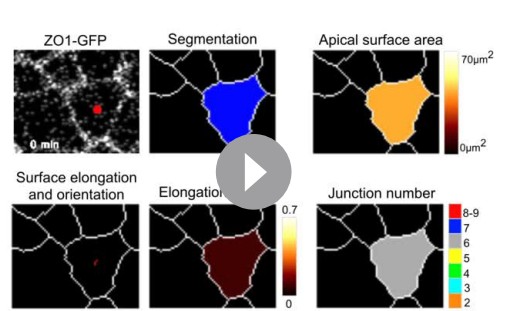

**Video 5.** Time-lapse of ZO-1-GFP highlighting a single cell at the primitive streak apically constricting. Membrane segmentation is performed over time such that different parameters can be followed and quantified, including the apical surface area, cell elongation and orientation, and the number of junctions: 95 min total with 5 min interval between frames.

https://elifesciences.org/articles/84019/figures#video5

Apical constriction has been extensively studied in *Drosophila*, including during mesoderm formation and neuroblast delamination (*An et al., 2017*; *Martin et al., 2009*; *Mason et al., 2013*; *Simões et al., 2017*). Very few studies have investigated the gastrulation EMT in an amniote, and it is unknown how cells constrict their apical surfaces during single-cell ingression in the mouse embryo versus the more global tissue-level constriction and invagination during *Drosophila* gastrulation. To follow changes in apical surfaces during cell ingression, cell membranes identified on projections of Z-stacks from time-lapse data were segmented using Tissue Analyzer software (*Aigouy et al., 2010*; *Aigouy et al., 2016*). By visualizing cells at the primitive streak, we identified apical constriction events associated with cell ingression and quantified several parameters, including apical surface area, apical cell surface elongation and orientation, and number of apical junctions (*Figure 1G*, *Figure 2—figure supplement 1*, *Videos 4 and 5*). Epiblast cells exhibited a variety of apical surface sizes and shapes that fluctuated over time as they flowed toward the primitive streak. As they initiated apical constriction at the primitive streak, cells exhibited 20–80 µm² apical surface area, and completed ingression within 25–90 min (*Figure 2A* and *Figure 2—figure supplement 1D*). With a 5 min image data acquisition time interval, the majority of ingressing cells exhibited pulsed constrictions, generally consisting of two to four major phases of contraction (*Figure 2B and C* and *Figure 2—figure supplement 1G*). As cells progressed toward ingression, the number of apical junctions decreased (*Figure 2—figure supplement 1J*, *Video 5*). We observed a similar behavior of pulsed apical constrictions with both the ZO-1-GFP and a myosin heavy chain IIB-GFP reporter (*Figure 2—figure supplement 1A–C*).

A closer examination of individual cells allowed us to quantify several features of apical surface area throughout the constriction period. The example in *Figure 2D* depicts the apical surface area of a cell exhibiting three main pulses of constriction each separated by more stable intermediate steps (*Figure 2C and D*, *Video 6*). While cells showed an average rate of constriction of 1.1±0.1 µm²/min (mean ± s.e.m., n=51 cells, four embryos), they exhibited constriction pulses with higher rates of contraction (up to 4.6 µm²/min), and an average pulse magnitude of 1.4±0.1 µm²/min (*Figure 2B, C and E* and *Figure 2—figure supplement 1H*). Cells at the primitive streak fluctuated between expansion and contraction before ingression, as did epiblast cells located some distance from the streak (*Figure 2C* and *Figure 2—figure supplement 1D, I*). During ingression, cells spent 90% of the time contracting apical surfaces (compare with 48% and 55% for epiblast cells away from the streak and before ingression) with an average rate of contraction of 1.1 µm²/min, higher than the average expansion rate of 0.1 µm²/min (*Figure 2E* and *Figure 2—figure supplement 1F*). To conclude, epiblast cells exhibited fluctuations in their apical surface areas before ingression, and pulsed constrictions during ingression at the primitive streak. We quantified the elongation of the apical surface, its orientation and the number of edges per cell at the onset and throughout the constriction in order to identify any parameters that could predict and identify cells that will initiate constriction and ingression, but no significant trend of hallmark were observed (*Figure 2—figure supplement 1J–L*).

## Asynchronous apical constriction at multi-cellular level

To uncover details of the apical constriction behavior at the multi-cellular level, we analyzed clusters of cells at the primitive streak. Segmenting cell clusters allowed tracking of cells and documentation of changes in their apical surfaces through time, as cells ingressed (*Figure 2F*, *Figure 2—figure supplement 2*, *Video 7*). Cells underwent fluctuations in their apical surface area, as some cells constricted and ingressed (*Figure 2—figure supplement 2B*, *Video 7*). By color-coding and plotting the rate of apical surface area change, we observed asynchronous oscillation of expansion and contraction phases among neighboring cells. As cells constricted and ingressed, their neighbors did not always exhibit the same behavior; some contracted at the same time, while others slightly expanded or maintained a constant surface area (*Figure 2F*, asterisk shows ingression; *Figure 2—figure supplement 2C* and *Video 7*). Thus, neighboring epiblast cells undergoing EMT and ingressing at the primitive streak displayed asynchronous apical behaviors and did not simultaneously ingress, with some neighboring cells ingressing 2 hr apart, in comparison to *Drosophila* gastrulation where all ventral furrow cells coordinately constrict in a time frame of less than 10 min.

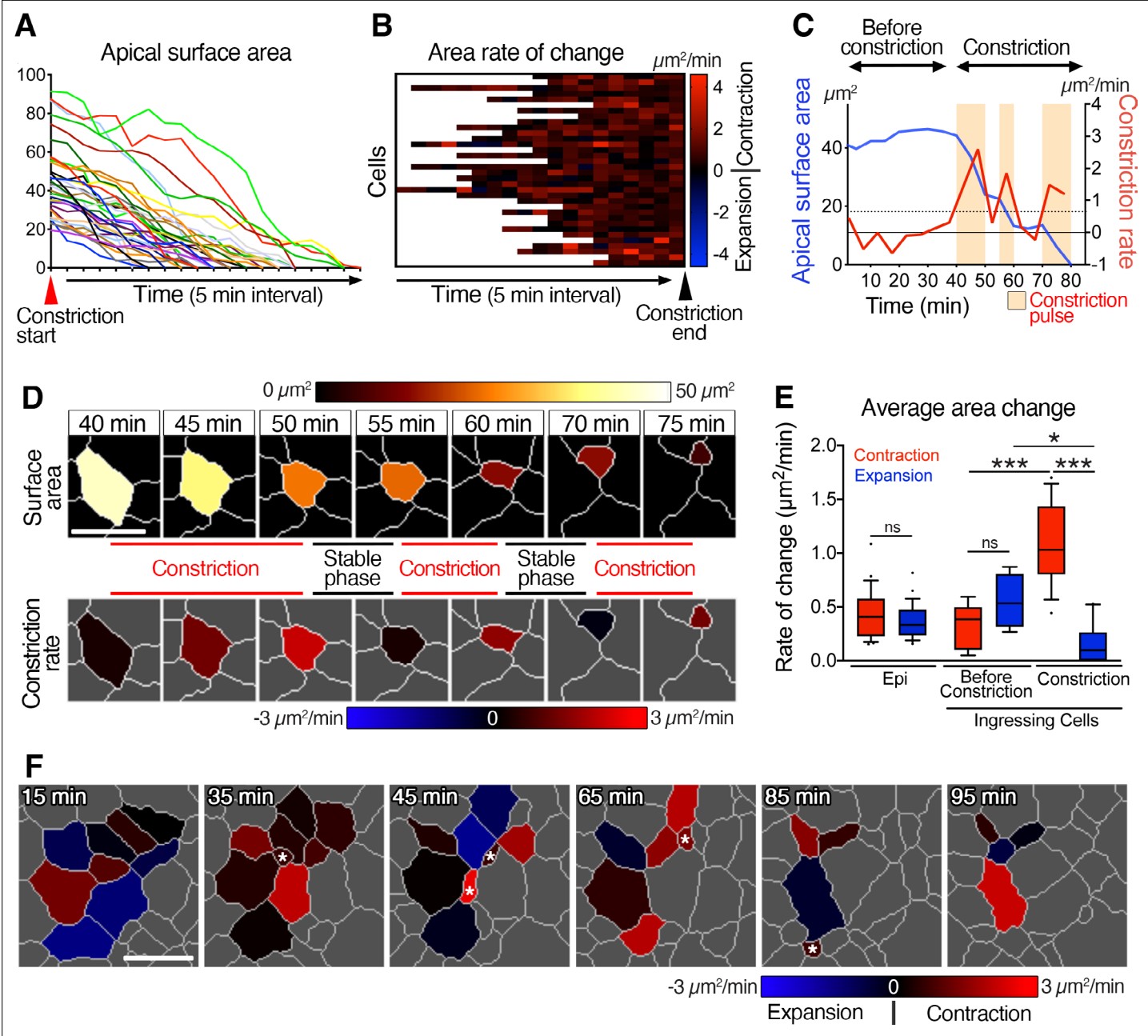

**Figure 2.** Pulsed racket-like apical constriction during epiblast cell ingression and asynchronous constriction at the multi-cellular level. (**A**) Graph showing the apical surface area of a few cells during constriction preceding ingression. Data are aligned to the start of constriction. (**B**) Heat map showing the rate of change of apical surface area of cells during the final constriction period, revealing minimal expansion and pulsed contractions. Each row represents data for an individual cell, with all data aligned to the end of the constriction period. (**C**) Graph of apical surface area and rate of constriction of a single cell exhibiting a slight oscillation in its apical surface before undergoing a final constriction period with (three – beige regions) pulses of constriction. (**D**) Membrane segmentation and color-coded time-series of apical surface area and constriction rate of a single cell exhibiting three pulses of constriction separated by periods of stability. The highlighted cell corresponds to the plot shown in C. (**E**) Plot showing the average rate of changes in apical area. Epiblast cells located at a distance from the primitive streak before ingression show low and equivalent levels of contraction and expansion. Cells at the primitive streak show no significant differences between contraction and expansion prior to their period of constriction, whereas rate of contraction is significantly higher, and expansion reduces during the subsequent constriction period. (**F**) Color-coded time-series and heat map of rate of change of apical surface area showing asynchronous oscillation in area and asynchronous constriction and ingression among neighbors. Asterisks show ingression events. Forty-one cells quantified from four embryos. Error bars represent s.e.m. See *Supplementary file 1* for n and p values. Scale bars: 10μm. *p<0.05, ***p<0.001 (unpaired bilateral Mann-Whitney test).

The online version of this article includes the following source data and figure supplement(s) for figure 2:

**Source data 1.** Apical constriction and constriction rate.

**Figure supplement 1.** Details and features of constriction pulses and cell parameters during constriction.

**Figure supplement 2.** Asynchronous apical constriction at the multi-cellular level.

## Asynchronous shrinkage of junctions correlates with apical constriction during cell ingression

Most apical junctions of any given cell decreased in length in a pulsed manner, with one to three pulses of shrinkage (*Figure 3A and B*), with junctions appearing to shrink asynchronously (*Figure 3A*, *Video 6*). By analyzing and quantifying the behavior of each junction of any one cell during constriction, we noted that the average rate of shrinkage was significantly higher during cell constriction pulses (*Figure 3C*). Junctions of individual cells shrunk asynchronously (*Figure 3D–F and H* and *Figure 3—figure supplement 1A–C, E, F*), differentially and with different magnitudes between one constriction pulse and the next (*Figure 3G* and *Figure 2—figure supplement 1D*), suggesting that asynchronous differential junctional shrinkage may trigger constriction of the cell surface. In sum, epiblast cells at the mouse primitive streak constrict their apical surfaces in a ratchet-like manner, with associated asynchronous shrinkage of junctions (*Figure 3F*).

## Anisotropic accumulation of components at apical junctions of epiblast cells

We sought to analyze the localization of apical proteins implicated in epithelial apical constriction in different tissues and species, that could potentially be involved in the apical constriction associated with the mouse gastrulation EMT. E7.5 embryos were immunostained, microdissected and the posterior region mounted so that the apical surface of epiblast cells could be positioned adjacent to the objective, and imaged in the vicinity of the middle of the primitive streak, which has the least curvature and is preferable for such an analysis (*Figure 4A* and *Figure 4—figure supplement 1A*). Crumbs2 and myosin heavy chain IIB have been described as exhibiting heterogenous and reciprocal enrichment at the apical junctions of epiblast cells (*Ramkumar et al., 2016*), and our data showed myosin heavy chain IIB and Crumbs2 to be predominantly anisotropically distributed, with Crumbs2 exhibiting a stronger anisotropy, being almost absent from some junctions, while myosin heavy chain IIB was observed on all junctions but at varying levels (*Figure 4D* and *Figure 4—figure supplement 1B*).

We next investigated the localization of other apical proteins, including the actomyosin components, mono- and di-phosphorylated myosin regulatory light chain (pMRLC and ppMRLC), Rock1, a Rho-associated kinase that phosphorylates and activates myosin regulatory light chain, and F-actin. We analyzed the localization of the Crumbs complex proteins PatJ, and aPKC, a

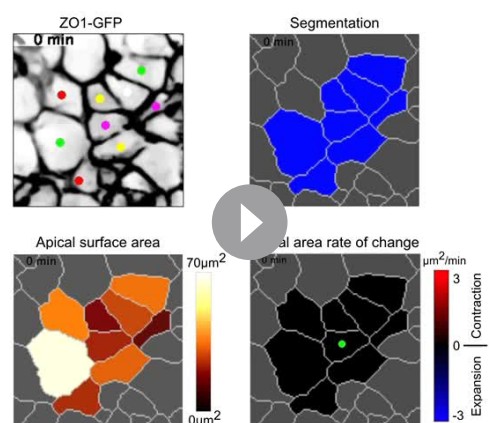

**Video 7.** Time-lapse of ZO-1-GFP showing a cell cluster at the primitive streak. Most of the cells gradually constrict and ingress, and cells tracked after segmentation and color-coded for apical surface area and changes in surface area. Images were denoised and deconvolved: 115 min with 5 min interval between frames.

https://elifesciences.org/articles/84019/figures#video7

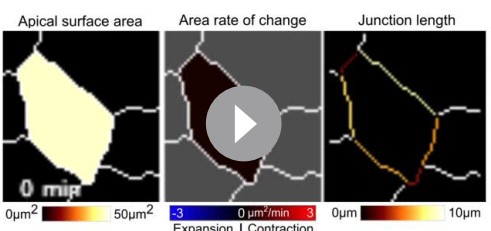

**Video 6.** Segmentation of time-lapse showing color code of cell apical surface (left panel), constriction rate (middle panel), and junction length (right panel) of a cell shown in *Figure 2*: 40 min total with 5 min interval between frames.

https://elifesciences.org/articles/84019/figures#video6

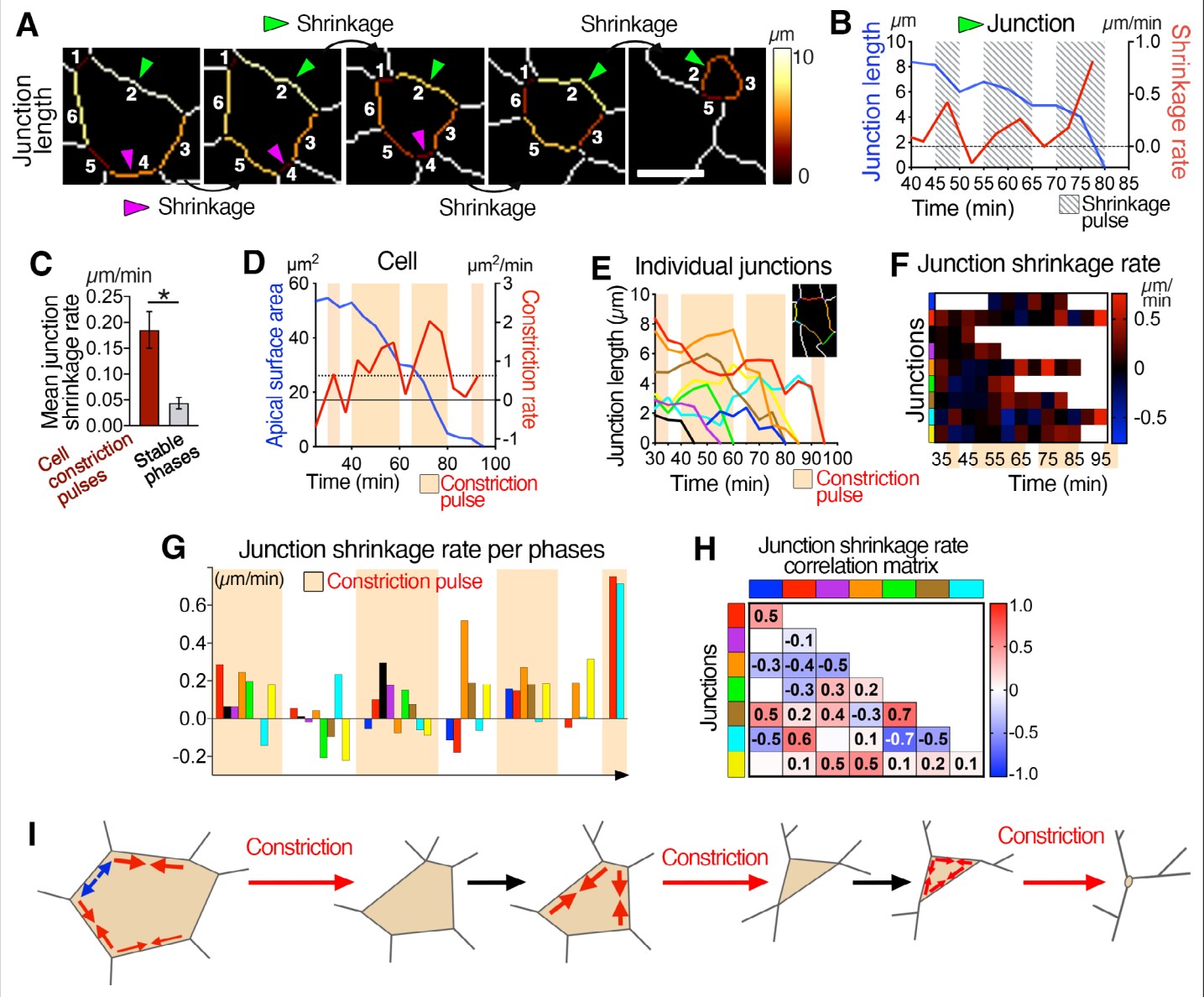

**Figure 3.** Asynchronous junctional shrinkage during apical constriction. (**A**) Membrane segmentation and color-coded time-series of a single junction showing a reduction of the junction's length during apical constriction. Here, the six initial junctions are numbered, and two are highlighted (arrowheads) and shrink at different time. (**B**) Graph showing the length and rate of shrinkage of the junction identified with the green arrowhead in A, indicating the three main pulses of shrinkage (hatched regions). (**C**) Graph showing increased rate of junctional shrinkage during the pulses of cell constriction compared to stable phases between constriction pulses from multiple cells (three embryos, 51 junctions). (**D**) Graph of apical surface area and rate of constriction of a single cell showing pulses of constriction, and associated graph showing the length of its individual junctions over time (**E**). Junctions reduce their length differentially during the constriction pulses (beige regions). (**F**) Heat map of the shrinkage rate of junctions, showing junctions shrinking asynchronously during the constriction pulses (beige time points). Each row represents data for an individual junction, junctions are color-coded as corresponding in E, and ordered as neighboring junctions. (**G**) Plot of the average shrinkage rate of each junction over time during each constriction pulse and intervening stable phases, illustrating the asynchronous shrinkage of different sets of junctions with variable magnitude during the different cell constriction pulses. Junctions are color-coded as neighbors, as in E. (**H**) Matrix heat map showing the correlation of junction length changes over time. Some junctions show correlation and change their length, as others show anti-correlation. Each row and column represent a single junction and are color-coded as in E. Numbers represent the correlation coefficients for each pair of junctions. (**I**) Model of ratchet-like pulsed constriction associated with asynchronous junctional shrinkage. Error bars represent s.e.m. *p<0.05 (unpaired bilateral Mann-Whitney test). See *Supplementary file 1* for n and p values. Scale bars, 5μm.

The online version of this article includes the following source data and figure supplement(s) for figure 3:

**Source data 1.** Junction length and shrinkage rate.

**Figure supplement 1.** Further example illustrating the asynchronous junctional shrinkage during apical constriction.

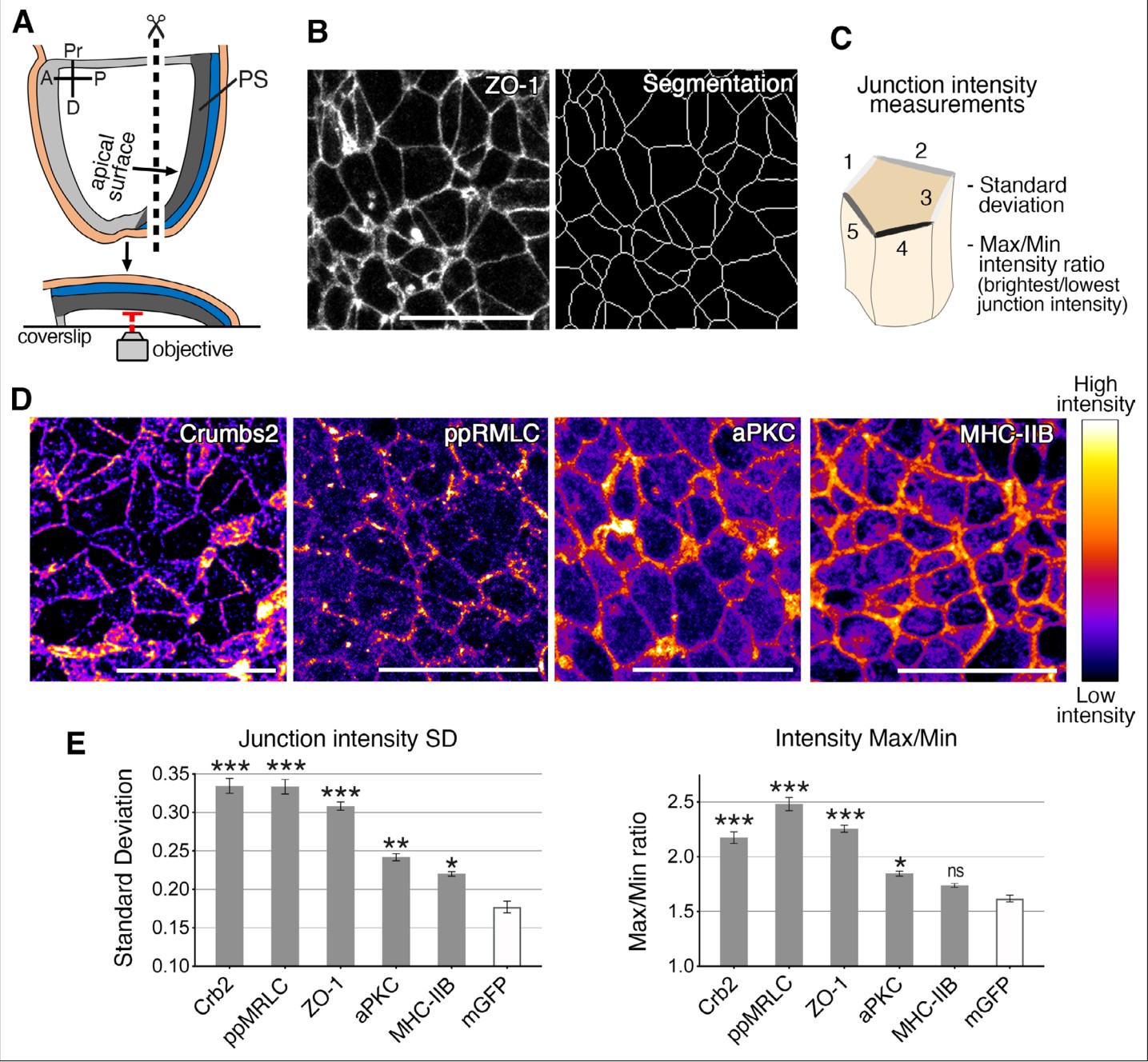

**Figure 4.** Anisotropic distribution of apical components in epiblast cells at the primitive streak. (**A**) Schematic of embryo processed for fixed tissue imaging after immunostaining. Embryos are microdissected and their posterior side is mounted between a slide and coverslip so that the apical surface of the epiblast is directly adjacent to the objective for imaging. (**B**) ZO-1 apical membrane localization, and corresponding skeleton of membrane segmentation used to quantify junctional intensities and anisotropy parameters, as schematized. (**C**) The average intensities of proteins associated with the junctions of individual cells are used to calculate the standard deviation between junctions, and the difference between junctions with the maximum and minimum intensities. (**D**) Lookup table (LUT) of immunostaining of Crumbs2, di-phosporylated myosin regulatory light chain (ppMRLC), aPKC, and myosin heavy chain IIB. These proteins show different distributions at apical junctions. Myosin heavy chain IIB and aPKC are present on almost every junction, and have a low anisotropy distribution, whereas ppMRLC and Crumbs2, which are not present on every junction, show a more anisotropic distribution. (**E**) Plots showing the anisotropy parameters, standard deviation, and Max/Min. Proteins such as Crumbs2, ZO-1, and ppMRLC show elevated parameters corresponding to a high level of anisotropy, whereas aPKC and myosin heavy chain IIB show lower anisotropy. The membrane-GFP (recombined mTmG mouse reporter line) is used as a control as it exhibits a homogeneous distribution of GFP across all junctions with low anisotropy. A range of 407–4135 junctions were analyzed in three to five embryos for each protein. Statistical tests compare each protein to GFP. Pr: proximal, D:

*Figure 4 continued on next page*

*Figure 4 continued*

distal, R: right, L: left, PS: primitive streak. \**p<0.01, \*\*\*p<0.001 (one-way ANOVA test). Error bars represent s.e.m. See *Supplementary file 1* for n and p values. Scale bars, 20μm.

The online version of this article includes the following source data and figure supplement(s) for figure 4:

**Source data 1.** Standard deviation and Max/Min ratio of the junctional intensity of the different proteins.

**Figure supplement 1.** Reciprocal enrichment of Crumbs2 and myosin heavy chain IIB, and anisotropy of apical components at the primitive streak.

**Figure supplement 2.** Apical proteins show anisotropy accumulation but no planar polarity.

regulator of non-muscle myosin II in different contexts and tissues (*Biehler et al., 2021*; *Petrov et al., 2017*; *Röper, 2012*; *Sidor et al., 2020*). Membrane segmentation allowed us to systematically quantify several parameters, including apical surface area, cell elongation, and the intensity of junctional protein accumulation (*Figure 4B*, *Figure 4—figure supplement 1C*). Two parameters were calculated based on junctional protein intensity and used to evaluate the level of anisotropy: the standard deviation of the junctional intensity measurements cells (SD), and the ratio of maximum to minimum junctional intensity values (Max/Min) for each cell (*Figure 4C*). The membrane-GFP reporter served as a control for homogeneous membrane fluorescence localization with low anisotropy, and the cohort of proteins analyzed exhibited varying degrees of anisotropy (*Figure 4D and E* and *Figure 4—figure supplement 1D, E*). Some proteins, including Crumbs2, ppMRLC, ROCK1, and ZO-1 showed a high level of anisotropy (*Figure 4D and E* and *Figure 4—figure supplement 1D, E*), though each exhibited a distinct pattern. ZO-1 was observed on all junctions, with variable intensity. Crumbs2 was present on some junctions but absent on others. ppMRLC and ROCK1 also accumulated on some junctions but were reduced on others. Other proteins, including PatJ and F-actin, displayed distinct patterns, but intermediate anisotropy. Finally, a subset of the proteins analyzed, including pMRLC, myosin heavy chain IIB, and aPKC, showed less anisotropy with lower SD and Max/Min values (*Figure 4D and E* and *Figure 4—figure supplement 1D, E*). These data allowed us to classify these apical proteins based on their junctional distribution and level of anisotropy. Notably, when analyzed at earlier (E7) and later stages (E8) of embryo development, these protein distributions did not exhibit visible differences (*Ramkumar et al., 2016*), arguing for a defined window of time where dynamic anisotropies could potentially facilitate a specific mode of cell ingression at gastrulation.

The anisotropic membrane accumulation of proteins is relevant with respect to the planar cell polarity (PCP) of epithelial cells. The apical proteins we analyzed did not show an overt planar polarized distribution, which we sought to verify quantitatively. We compared the planar polarity index calculated by Tissue Analyzer (amplitude and orientation of polarity) and mean junctional intensity in relation to junction orientation (*Figure 4—figure supplement 2*) in two different ways, to assess the planar polarity of apical proteins as compared to Celsr1, a protein with a planar polarized distribution implicated in PCP in mouse and chick (*Curtin et al., 2003*; *Mahaffey et al., 2013*; *Nishimura et al., 2012*). Celsr1 exhibited a clear planar polarized localization and tended to accumulate on junctions oriented on left-right axis perpendicular to the proximo-distal axis. By contrast, myosin heavy chain IIB, Crumbs2, ppMRLC, ZO-1, and aPKC exhibited no specific axis of planar polarity. These results show that actomyosin network and Crumbs2 complex proteins at the apical junctions do not exhibit planar polarity, but a disordered anisotropy distribution.

## Reciprocal distribution of apical proteins at apical junctions

We next assessed the colocalization and correlation of these different apical proteins at junctions. Segmentation and intensity values, which served as a readout of concentration, were used to systematically compare mean junctional protein intensities and correlate localization. Some proteins such as ppMRLC and Rock1 showed similar patterns with comparable levels of anisotropy and appeared to accumulate on the same junctions (*Figure 5A*). Co-staining revealed a broader accumulation for ppMRLC. However, where Rock1 was seen to accumulate, ppMRLC was always present, demonstrating strong colocalization (*Figure 5A and C*), close to that observed with a positive control staining for ppMRLC with two different secondary antibodies (*Figure 5C* and *Figure 5—figure supplement 1C, E*). F-actin accumulated on most junctions, exhibiting a distinct and broader distribution compared to ppMRLC. However, their intensities exhibited a high level of colocalization (*Figure 5—figure supplement 1A*), with high correlation at junctions (*Figure 5C* and *Figure 5—figure supplement 1A, E*).

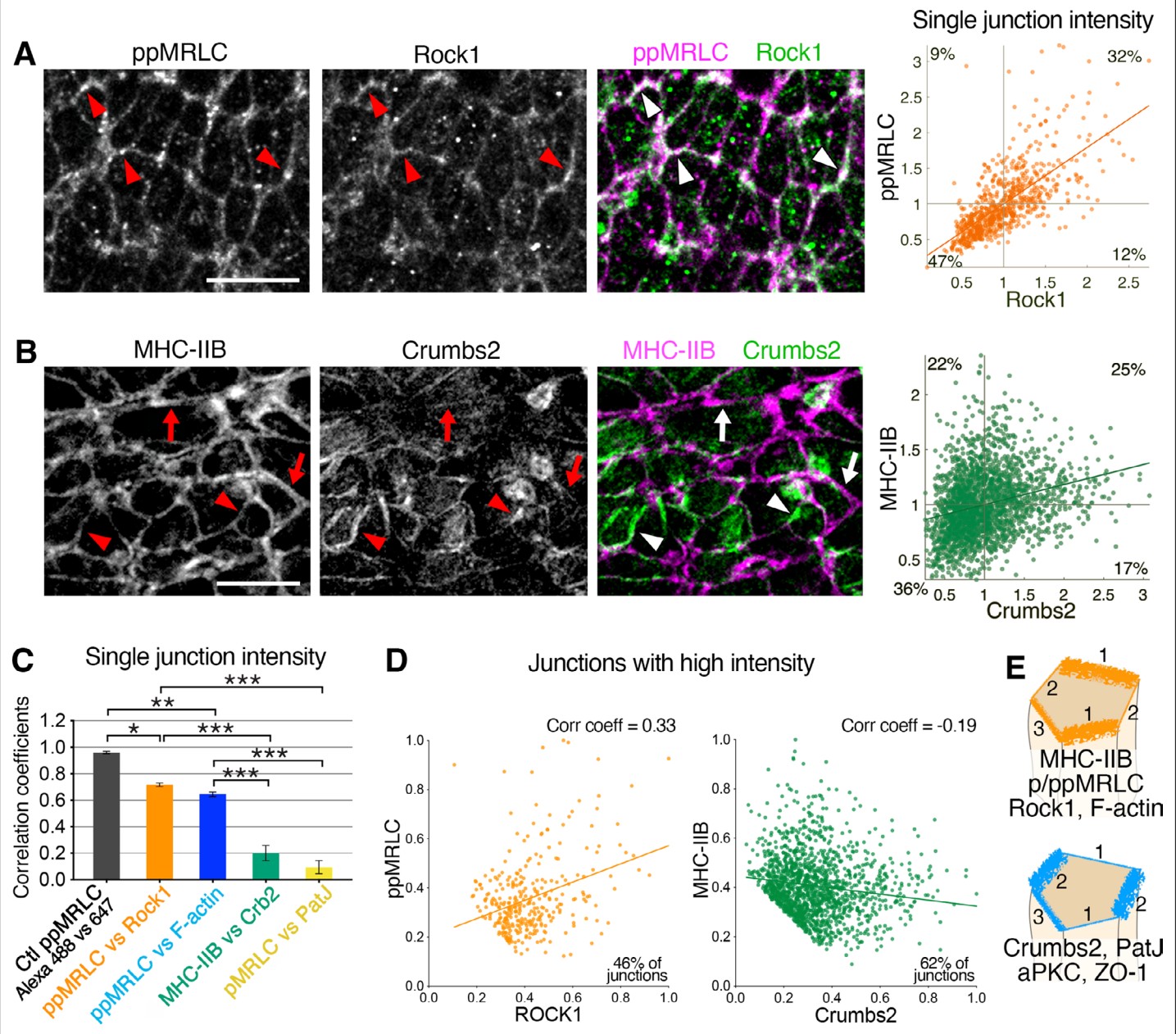

**Figure 5.** Reciprocal enrichment of apical components at apical junctions. (**A**) Co-immunostaining of di-phosphorylated myosin regulatory light chain (ppMRLC) and Rock1 reveals their tendency to accumulate on the same junctions. Note colocalization of fluorescent signal on most junctions (arrowheads). Scatter plot on the right illustrating positive intensity correlation of the two proteins. Each point represents the mean intensity at a single junction. (**B**) Co-immunostaining of myosin heavy chain IIB and Crumbs2 reveals that they tend to accumulate on different junctions. Note junctions where myosin heavy chain IIB accumulation predominates (arrows), and junctions where Crumbs2 predominates (arrowheads). Corresponding scatter plot shows a low correlation. (**C**) Plot showing the correlation coefficients associated with the different scatter plots. Proteins such as ppMRLC, Rock1, and F-actin exhibit a strong correlation with a high coefficient (ppMRLC vs Rock1=0.72 ± 0.01, ppMRLC vs F-actin=0.65 ± 0.02), whereas others including myosin heavy chain IIB and Crumbs2, PatJ, and pMRLC exhibit much lower correlation with low coefficient (myosin heavy chain IIB vs Crumbs2=0.20 ± 0.06, pMRLC vs PatJ = 0.09 ± 0.05). (**D**) Scatter plot showing the junctional intensity correlation of ppMRLC and Rock1; and Crumbs2 and myosin heavy chain IIB where junctions with low intensity were removed from the analysis (intensities were normalized from 0 to 1). ppMRLC and Rock1 continue to show a positive correlation as in A, while myosin heavy chain IIB and Crumbs2 show anti-correlation. (**E**) Based on co-staining for multiple apical proteins and measurements of anisotropy and correlation, two groups of proteins can be defined with a tendency to accumulate on same junctions: actomyosin components, and apical determinants and junctional proteins. Junction 1 predominantly accumulates the myosin group, with very little accumulation of the Crumbs group. Junction 2, on the other hand, predominantly accumulates the Crumbs group with very little accumulation of the myosin group.

*Figure 5 continued on next page*

Figure 5 continued

While junction 3 shows an intermediate accumulation of both groups of proteins; 711–2453 junctions were quantified in three to four embryos for each pair of proteins. *p<0.05, ***p<0.001 (one-way ANOVA test). Error bars represent s.e.m. See *Supplementary file 1* for n and p values. Scale bars, 10 µm.

The online version of this article includes the following source data and figure supplement(s) for figure 5:

**Source data 1.** Mean junctional intensity of the different proteins.

**Figure supplement 1.** Correlation of protein distributions at apical junctions.

**Figure supplement 2.** Correlation of protein distributions at junctions of high intensity.

**Figure supplement 3.** Junctional correlation of proteins at the single cell level.

Comparison of F-actin and Rock1 revealed a similar high correlation (*Figure 5—figure supplement 1D, E*). Thus, the correlated distributions of ppMRLC, Rock1, and F-actin proteins suggest they could function at the same junctions. Despite exhibiting anisotropy, ZO-1 and aPKC were observed to be present on all junctions (*Figures 4B, D, E , and 3*), with a high level of correlation, similar to a positive control (*Figure 5—figure supplement 1D, E*), demonstrating their accumulation on the same junctions.

By contrast, when comparing other junctional proteins, we observed very low or no correlation. Crumbs2 and myosin heavy chain IIB exhibited a reciprocal enrichment (*Figure 5B*), with minimal colocalization. Comparisons of Crumbs2 and myosin heavy chain IIB junctional intensities revealed a broad distribution and very low correlation (*Figure 5B and C*). Comparing PatJ, a Crumbs partner protein, with pMRLC showed a similar accumulation with almost no correlation (*Figure 5C* and *Figure 5—figure supplement 1B, E*). Analysis of other protein pairs, including ZO-1-ppMRLC, and aPKC-ppMRLC, revealed low correlation (*Figure 5—figure supplement 1D, E*).

Some proteins such as myosin heavy chain IIB and Crumbs2, and PatJ and pMRLC show visible reciprocal enrichment on immunostaining, while plotting junction intensities shows no correlation (*Figure 5B* and *Figure 5—figure supplement 1B*). We removed the junctions with low intensity from the analysis and keep the ones with high intensities where reciprocal enrichment mainly happens. When these selected junctions were plotted, several pairs of proteins (myosin heavy chain IIB and Crumbs2, aPKC and ppMRLC, PatJ and pMRLC, ZO-1, and ppMRLC) now showed anti-correlation (correlation coefficients from –0.16 to –0.19) (*Figure 5D* and *Figure 5—figure supplement 2A, B*), suggesting their reciprocal enrichment. Other pairs of proteins continue to show a positive correlation (correlation coefficients from +0.33 to +0.91). We also analyzed junctional intensity correlations grouped by individual cells, rather than pooled for all cells. This analysis confirmed the strong correlation of ppMRLC, Rock1, and F-actin, where most cells exhibited a positive correlation trend with a high correlation coefficient (*Figure 5—figure supplement 3A, B*). For pairs of proteins such as Crumbs2 and myosin heavy chain IIB, and PatJ and pMRLC, grouping junction intensities by individual cells revealed a broad distribution, some cells showed anti-correlation, some no correlation and some positive correlation (*Figure 5—figure supplement 3C–F*). This analysis illustrated a heterogeneity of proteins across the epiblast cell population in the vicinity of the primitive streak, at any one time. We hypothesize that these anisotropic accumulation patterns are likely to be dynamic.

These quantitative analyses of apical protein anisotropy patterns combined with the correlation of distributions allowed us to define two groups of proteins that tend to accumulate on different junctions. First, cytoskeletal proteins including myosin heavy chain IIB, pMRLC, ROCK1, and F-actin predominantly accumulate on the same junctions (*Figure 5E*, junctions 1), and second, apical components and junctional proteins such Crumbs2, PatJ, aPKC, ZO-1 predominantly colocalize on other junctions (*Figure 5E*, junctions 2). These two groups of proteins can also show intermediate accumulation on the same junctions, which we hypothesize to be a transient state, as we believe these anisotropic patterns to be dynamic in time. This anisotropic reciprocal enrichment of the actomyosin network and Crumbs complex-associated proteins is associated with dynamic apical constriction.

## Cellular defects in *Crb2*^-/- mutant embryos at the primitive streak

Crumbs2 is an important regulator of the mouse gastrulation EMT, where it promotes cell ingression. In *Crb2*^-/- embryos, an early wave of EMT takes place, producing a small number of mesodermal cells (*Ramkumar et al., 2016*). Thus, Crumbs2 appears not to be required at the onset of gastrulation, as mutants exhibit major defects in cell ingression at later stages. Defects are evident by E7.75 when

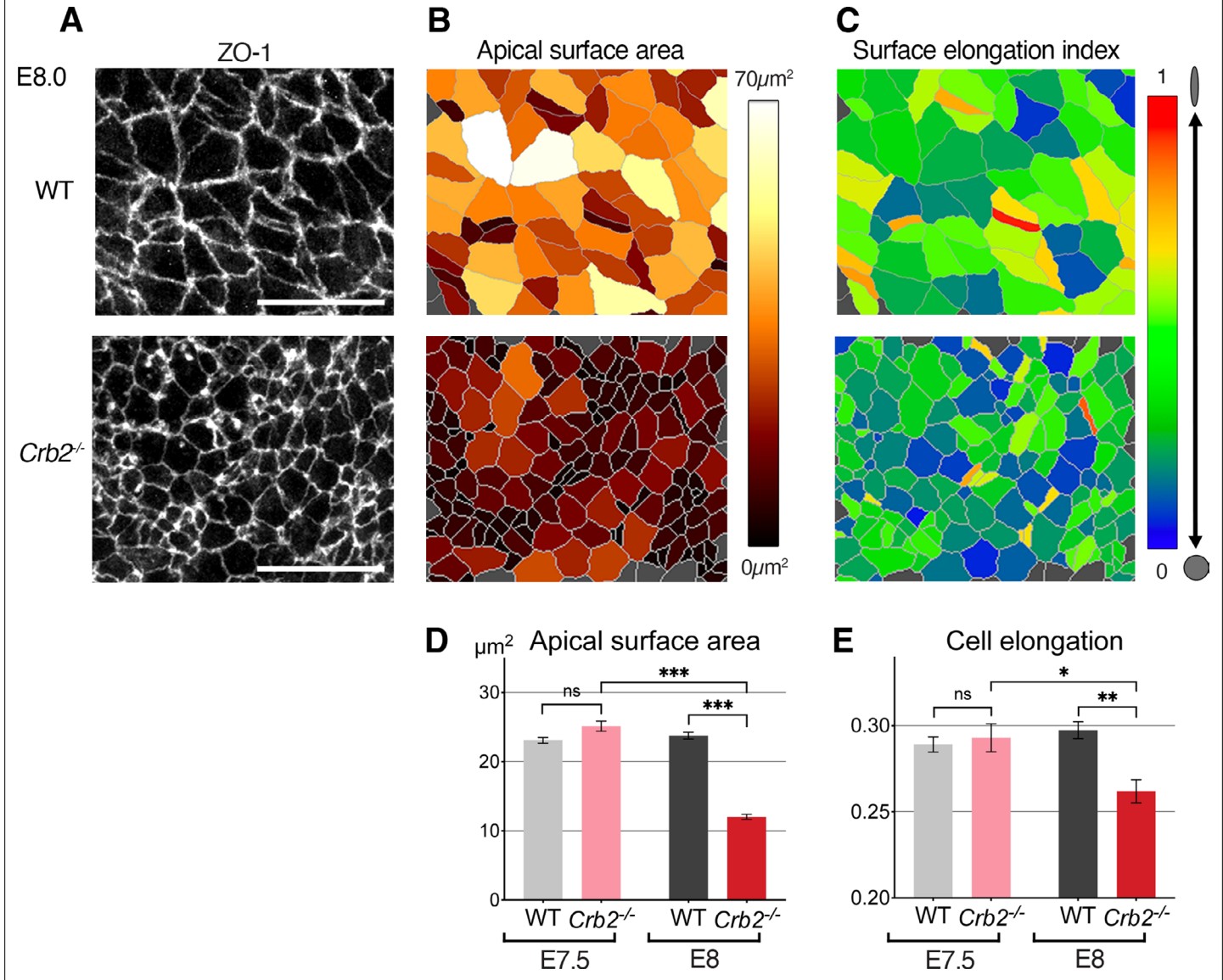

**Figure 6.** Cellular defects in *Crb2* mutant embryos at the primitive streak. (**A**) ZO-1 immunostaining of the apical surface of wild-type (WT) and *Crb2* mutant embryos at embryonic day (E)7.5 and E8.0. (**B**) Segmentation and color code of apical surface areas reveals smaller surfaces in *Crb2* mutants at E8.0. (**C**) Color code of cell surface elongation index calculated by Tissue Analyzer (regardless of their axis) reveals rounder surfaces in *Crb2* mutants at E8.0. (**D**) Graph of quantifications showing significantly smaller apical surfaces in *Crb2* mutants at E8.0 compared to E7.5, and to WT embryos. (**E**) Graph showing significantly less elongated apical surfaces in *Crb2* mutants at E8.0 compared to E7.5, and to WT embryos; 334–1187 cells from three to five embryos were quantified for each genotype and stage. ***p<0.001 (unpaired bilateral Mann-Whitney test). Error bars represent s.e.m. See *Supplementary file 1* for n and p values. Scale bars, 20 µm.

The online version of this article includes the following source data and figure supplement(s) for figure 6:

**Source data 1.** Apical surface area and cell elongation in wild-type (WT) and *Crb2* mutant embryos.

**Figure supplement 1.** Cellular and molecular defects in *Crb2* mutant embryos.

epiblast cells fail to ingress and accumulate at the primitive streak. By quantifying the apical surface of posterior epiblast cells at E8.0, we observed an increased cell density and cells with smaller apical surfaces around the primitive streak of mutants, as compared to littermate controls (*Figure 6—figure supplement 1A*). This cell crowding was not observed 12 hr earlier at E7.5 (*Figure 6—figure supplement 1B*; *Ramkumar et al., 2016*) when mutant embryos were morphologically indistinguishable from wild-type (WT) stage-matched counterparts. Morphological defects (apical surface areas and surface elongation) in cells at the primitive streak were not observed at E7.5 and were first detected at

E8.0 with cells having smaller and rounder apical surfaces (*Figure 6A–E* and *Figure 6—figure supplement 1A, B*) resulting from their failure to ingress, and consequent accumulation in the epiblast.

## Molecular defects in *Crb2*[-/-] mutant embryos precede cellular defects

To identify molecular defects associated with and preceding the ingression defect, we analyzed the distribution of actomyosin components in *Crb2*[-/-] embryos. By measuring the junctional intensities and normalizing to the medial signal, although the distribution of ZO-1 appeared unchanged in mutants, suggesting the integrity of tight junctions was not altered, we observed reduced junctional levels of myosin heavy chain IIB and ppMRLC (the fully activated form of myosin regulatory light chain) in mutants at E7.5 (*Figure 7A and B* and *Figure 6—figure supplement 1C*). We also observed more diffuse and reduced cortical actin filaments at the apical junctions of epiblast cells in mutants, whereas the medial pool of actin appeared slightly increased (*Figure 7A and B*). Thus, actomyosin apical accumulation and activity were perturbed without alteration of junctional integrity in *Crb2*[-/-] embryos at E7.5, preceding morphological defects.

We next focused on aPKC and Rock1, which have been implicated in the regulation of the actomyosin network downstream of Crumbs in *Drosophila* and cultured cells (*Biehler et al., 2021*; *Ishiuchi and Takeichi, 2011*; *Röper, 2012*; *Sidor et al., 2020*). By quantifying the intensity of aPKC and Rock1 at junctions and normalizing to the medial signal, we observed a reduction of aPKC intensity at apical junctions and a more diffuse signal in the cytoplasm, and we observed a reduction of junctional Rock1 (*Figure 7C* and *Figure 6—figure supplement 1D*). These observations suggest a network involving aPKC and Rock1 regulating actomyosin acting downstream of Crumbs2. By quantifying the two parameters used to assess the anisotropic distribution of proteins at apical junctions, we observed that in *Crb2*[-/-] embryos the SD and Max/Min of junction intensity of myosin heavy chain IIB, F-actin ZO-1, aPKC, and Rock1 was reduced (*Figure 7D*), revealing reduced anisotropy and a more homogenous distribution of these proteins.

## Discussion

To develop a dynamic mechanistic understanding of the mouse gastrulation EMT at cellular resolution and tissue-scale, we performed 3D time-lapse imaging of ex utero cultured embryos. We used genetically encoded membrane-tagged fluorescent reporters and quantified the spatiotemporal features of apical cell surfaces, junctions, and junctional composition of cells in the vicinity of the primitive streak, the site of the gastrulation EMT. Our analyses reveal that after converging at the primitive streak, epithelial epiblast cells undergo EMT, either individually or in small groups, across the domain of the primitive streak. Cells constrict their apical surfaces in a ratchet-like manner as they ingress from the epiblast epithelium. We also note that this constriction is associated with the anisotropic apical distribution of actomyosin network and Crumbs2 complex proteins.

Apical constriction is an evolutionarily conserved cellular behavior associated with gastrulation in invertebrates such as *Drosophila*, nematodes, and sea urchins, as well as in vertebrates and amniotes, such as *Xenopus* and chick (*Christodoulou and Skourides, 2015*; *Lecuit and Lenne, 2007*; *Martin and Goldstein, 2014*; *Martin et al., 2009*; *Rozbicki et al., 2015*). We observed fluctuations in the size and shape of the apical surface of epiblast cells as they converged toward the mouse primitive streak. In the vicinity of the streak, and as cells ingressed, their apical surfaces predominantly constricted, in a step-wise ratchet-like manner.

Pulsed ratchet-like apical constrictions have been observed during *Drosophila* gastrulation, where cells at the ventral furrow constrict in a short period of time to drive tissue invagination, followed by their en-masse delamination (*Martin et al., 2009*; *Mason et al., 2013*). *Drosophila* neuroblast delamination also exhibits a ratchet-like apical constriction (*An et al., 2017*; *Simões et al., 2017*). Despite possessing some similarities, the processes by which mesoderm forms during gastrulation in *Drosophila* and mouse are difficult to directly compare, as they exhibit different time-scales (~1 hr vs. >24 hr), mechanisms, and spatial parameters. In *Drosophila*, cells at the ventral furrow constrict over 9–10 min, triggering a tissue-wide invagination without cell ingression. Thereafter, cells lose their epithelial properties as they form mesoderm, a mesenchymal tissue. By contrast, in the mouse embryo epiblast cells at the primitive undergo ingression in isolation. Cells undergoing ingression constrict

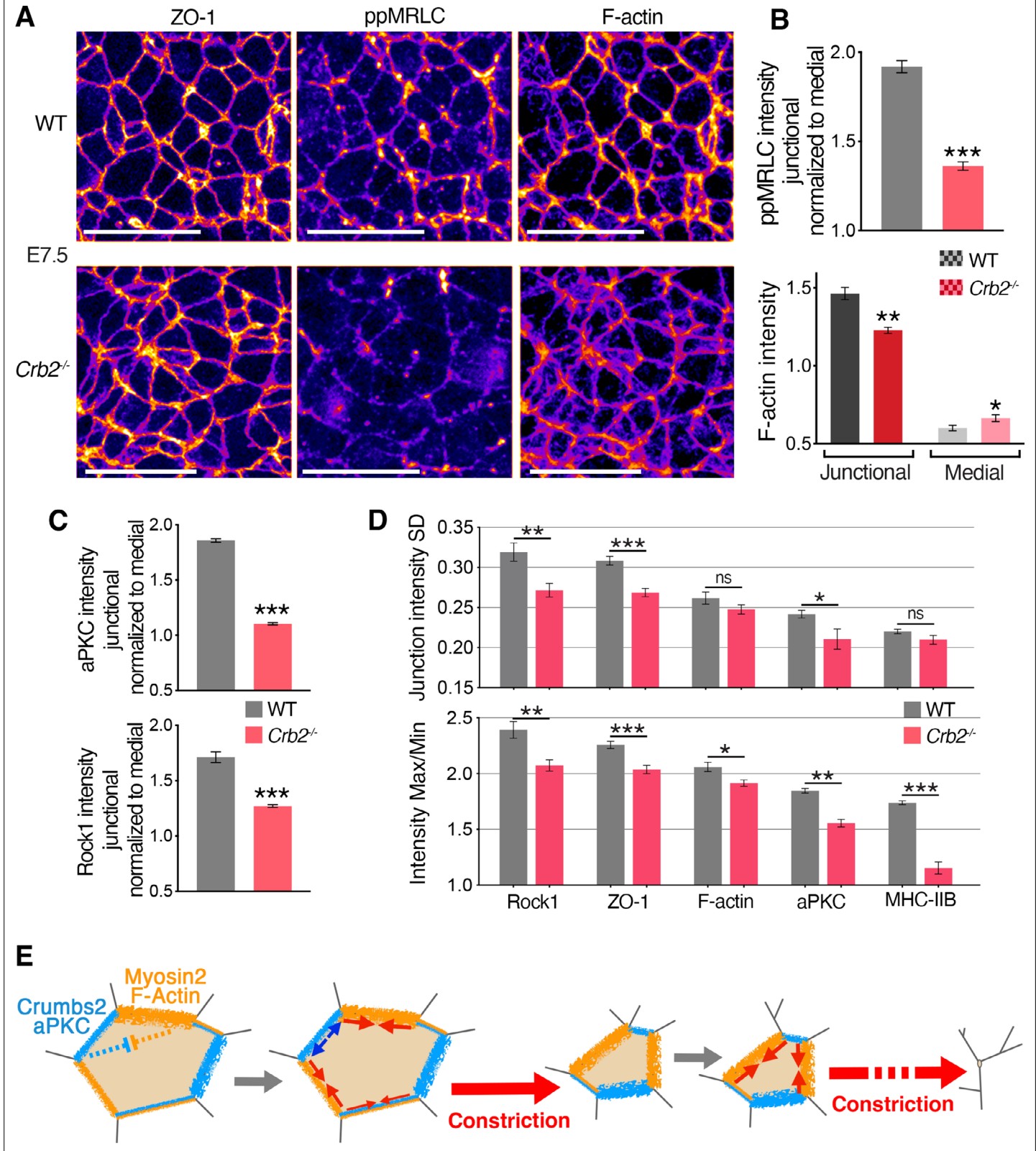

**Figure 7.** Molecular defects in *Crb2* mutants preceding visible cellular defects. (**A**) Co-immunostaining for ZO-1, di-phosphorylated myosin regulatory light chain (ppMRLC) and F-actin in wild-type (WT) and *Crb2* mutants at embryonic day (E)7.5 reveal reduced ppMRLC and F-actin junctional intensities. (**B**) Plots showing intensity measurements of junctional ppMRLC (junction intensities normalized by the medial signal) and F-actin. ppMRLC shows a significant decrease in junctional accumulation in *Crb2* mutants. F-actin shows lower junctional accumulation and increased diffuse medial intensity

*Figure 7 continued on next page*

*Figure 7 continued*

in *Crb2* mutants. (**C**) Graphs of junctional intensity measurements normalized by the medial signal, showing significantly junctional decreased aPKC and Rock1 intensity in *Crb2* mutants compared to WT. (**D**) Anisotropy parameters, standard deviation and Max/Min showing perturbed junctional accumulation anisotropy in *Crb2* mutants compared to WT embryos. ZO-1, F-actin, myosin heavy chain IIB, aPKC, and Rock1 show reduced anisotropy in *Crb2* mutants. (**E**) Working model. Actomyosin network and Crumbs2 complex show anisotropic and reciprocal enrichment at apical junctions of epiblast cells at the primitive streak. These distributions are likely dynamic through time and heterogenous among cells, and thus could potentially regulate each other's distribution. Junctions asynchronously shrink and allow a ratchet-like constriction of apical surfaces, thereby facilitating cell constriction and ingression without creating increased tension across the tissue or compromising its integrity; 122–501 cells in three to four embryos were quantified for each genotype and proteins. *p<0.05, **p<0.01, ***p<0.001 (B,C: unpaired bilateral Mann-Whitney test; D: one-way ANOVA test). Error bars represent s.e.m. See *Supplementary file 1* for n and p values. Scale bars, 20 μm.

The online version of this article includes the following source data for figure 7:

**Source data 1.** Intensity measurement of di-phosphorylated myosin regulatory light chain (ppMRLC), F-actin, aPKC, and Rock1 in wild-type (WT) and *Crb2* mutant embryos.

their apical surface over 25–90 min, a time-scale three to nine times longer than in *Drosophila*. Such differences in time-scale likely reflect similar but distinct mechanisms of constriction.

Live imaging of the apical surface of cells at the primitive streak in developing mouse embryos is technically challenging due to the epiblast being the most internal tissue layer of gastrula stage embryos. Even under optimized conditions, maintaining embryos viability and fluorescent signal intensity, the shortest time interval we could use to reliably collect data was 5 min. Given this time resolution, our time-lapse imaging revealed a ratchet-like apical cell constriction operating during the mouse gastrulation EMT, where cells predominantly ingress as isolated small clusters of cells, versus the en-masse constriction and invagination taking place during *Drosophila* gastrulation. Future studies where time-lapse imaging is performed with shorter time intervals should reveal further details of the constriction process.

The apical constriction we observed is associated with the asynchronous contraction of junctions between neighboring cells, and our fixed tissue imaging showed that it is potentially regulated by the apical actomyosin network. During *Drosophila* gastrulation, an apico-medial network of acto-myosin pulls apical cell junctions to simultaneously reduce their length and allow the constriction of the apical surface of cells at the ventral furrow (*Coravos and Martin, 2016*; *Martin et al., 2009*; *Mason et al., 2013*). *Drosophila* neuroblasts exhibit polarized apical constrictions driven by both planar polarized junctional and medial actomyosin networks, which predominantly lead to a loss of antero-posterior junctions (*Simões et al., 2017*). In the mouse gastrula, we found that cells undergoing EMT contract and lose cell-cell junctions in an asynchronous manner without apparent planar polarity. Our analyses of the junctional distribution of myosin II supports its potential involvement in regulating apical constriction during cell ingression, but does not exclude an additional role played by a medial pool. Pulsed actomyosin activity has been documented during various processes involving cell shape changes, including epithelial apical constriction (*Coravos et al., 2017*; *Heer and Martin, 2017*; *Martin and Goldstein, 2014*). The pulsed constriction we observe in mouse gastrulation indicates conservation of this actomyosin behavior as a general mechanism that integrates assembly and disassembly of the network and rearrangement and recruitment of myosin and actin molecules. Furthermore, pulsed contractility could allow a substantial but transient constriction of epiblast apical surfaces during ingression without creating deleterious tension which could impact tissue integrity.

Since a large numbers of cells undergo EMT and leave the epiblast epithelium at any given time, simultaneous constriction and ingression of multiple neighboring cells could exert extreme tension and deformation in the tissue. Our observation of scattered and asynchronous constriction and ingression of cells at the mouse primitive streak provides a parsimonious explanation for maintaining epithelial integrity, and a continuation of ingression events for over 24 hr throughout gastrulation, in comparison to *Drosophila* gastrulation where all ventral furrow cells form the mesoderm layer in approximately 1 hr. Cell division taking place on the basal side of the epiblast layer has been reported at the primitive streak where it has been proposed to facilitate cell ingression (*Mathiah et al., 2020*). Apical constriction and basal cell division could therefore work in concert to promote the correct number of cell ingression events.

Myosin II is a driver of membrane movement and apical constriction in various contexts (*An et al., 2017*; *Chung et al., 2017*; *Marston et al., 2016*; *Martin and Goldstein, 2014*; *Martin et al.,*

*2009*; *Mason et al., 2013*; *Nishimura et al., 2012*; *Roh-Johnson et al., 2012*; *Simões et al., 2017*). Crumbs2 is involved in the EMT process during mouse gastrulation and exhibits a reciprocal enrichment to myosin heavy chain IIB (*Ramkumar et al., 2016*; *Figure 5*). The anisotropic membrane localization of Crumbs has been implicated in *Drosophila* salivary gland development, where it controls the localization of actomyosin cables at the placode border, necessary for apical constriction and tissue invagination (*Röper, 2012*). Recently, Crumbs has also been implicated in apical constriction during neuroblast cell ingression in *Drosophila* (*Simões et al., 2022*). We observed the anisotropic and reciprocal enrichment of Crumbs2 and myosin heavy chain IIB, and have identified other proteins exhibiting anisotropic distributions. Although myosin heavy chain IIB, F-actin, aPKC, and ZO-1 are present on all apical junctions, they exhibit a differential anisotropic accumulations between the junctions of any one cell. Other proteins such as ppMRLC and the myosin kinase Rock1, as well as Crumbs2 and its partner PatJ exhibited a higher level of anisotropy. In the *Drosophila* salivary gland placode border, Crumbs and myosin-2 show planar polarized localization with Crumbs being absent from cell junctions that contact the border of the placode, allowing the accumulation of myosin-2 at the placode border and formation of a supracellular actomyosin cable (*Röper, 2012*). By contrast, our results in mouse show that Crumbs2, myosin heavy chain IIB, and several other proteins do not exhibit planar polarity, but instead exhibit a disordered anisotropy. Our studies reveal a patterned distribution in which actomyosin regulators and Crumbs2, aPKC, and ZO-1 accumulate on subsets of cell junctions where they display reciprocal enrichment. We hypothesize that these anisotropic distributions are dynamic, and serve to control the asynchronous junction contraction associated with apical constriction.

Crumbs acts through aPKC and Rock to regulate myosin II localization in the *Drosophila* salivary gland (*Röper, 2012*), and aPKC has been shown to regulate myosin II junctional localization and activity both positively and negatively in different context, directly or through other proteins (*Biehler et al., 2021*; *Ishiuchi and Takeichi, 2011*; *Petrov et al., 2017*). The regulation of myosin II by Crumbs and aPKC through Rock in *Drosophila* salivary gland development is complex (*Sidor et al., 2020*), and despite Rock membrane association being tightly regulated by Crumbs across the tissue, regionally distinct behaviors have been observed. At placode borders, cells exhibit a polarized accumulation of Crumbs/aPKC and Rock to control actomyosin cable formation, whereas in the center, where cells constrict during the tissue invagination, both Crumbs and Rock are isotropically present at junctions (*Chung et al., 2017*; *Sidor et al., 2020*). Our results demonstrate that during mouse gastrulation Crumbs2 controls the apical accumulation and activity of actomyosin, which could account for the defects in apical constriction and ingression in *Crb2* mutants. We observed Crumbs2 as required for the correct accumulation and distribution of aPKC, myosin heavy chain IIB, Rock1, and F-actin at apical junctions, without alteration of junctional integrity. Crumbs2 along with aPKC could control the anisotropic accumulation of actomyosin, triggering asynchronous junctional reductions resulting in apical constriction during ingression (*Figure 7E*). The defects observed in *Crb2* mutant embryos are complex; epiblast cells residing both at, and beyond, the primitive streak exhibit reduced apical surfaces (*Ramkumar et al., 2016*; *Figure 6—figure supplement 1A*). Since a large number of cells fail to ingress and remain within the epiblast layer of *Crb2* mutants, this results in cell crowding which could lead to changes in cell shape, including a reduction of apical surfaces.

Our work reveals that pulsed ratchet-like constriction behaviors observed in different species and distinct contexts are conserved in the mouse gastrulation EMT. Crumbs2 together with aPKC and Rock1 could be involved in a network required for myosin II localization and activity at the apical membrane of ingressing mouse epiblast cells. Crumbs2 can potentially also regulate the anisotropic accumulation of the actomyosin network locally at junctions of single cell, facilitating correct apical constriction during ingression (*Figure 7E*). The defects observed in *Crb2* mutants and the role of Crumbs2 in ingression appear very complex. Further studies will be necessary to determine why Crumbs2 is not required for cell ingression at the onset of gastrulation, and to glean its detailed mechanistic role in cell shape changes, notably apical constriction, but potentially also other membrane movements associated with cell ingression.

## Methods

### Experimental animals

FVB or mixed background mice were used for immunofluorescence staining. Most other experiments, incorporating genetically engineered mouse models, were carried out in a mixed background. The *Gt(ROSA)26Sor*^tm4(ACTB-tdTomato,-EGFP)Luo^/J (**Muzumdar et al., 2007**) (referred to throughout as *Rosa26*^mT/mG^) line was obtained from the Jackson Laboratory and crossed to Sox2-Cre (**Hayashi et al., 2002**) mice to generate a membrane-GFP reporter. The ZO-1-GFP protein fusion knock-in mouse reporter was a gift from Terry Lechler (Duke University Medical Center, NC, USA) (**Foote et al., 2013**; **Huebner et al., 2014**). The myosin heavy chain IIB-GFP protein fusion knock-in mouse reporter was a gift from Robert S. Adelstein (Laboratory of Molecular Cardiology, NHLBI, NIH, Bethesda, MD, USA) (**Bao et al., 2007**). *Crb2*^+/-^ were generated from animals carrying a conditional allele of *Crb2* via germline deletion (**Alves et al., 2013**). Animals were housed and bred in accordance with institutional IACUC guidelines. The MSKCC IACUC approved all experiments.

### Embryo dissection, immunostaining, and antibodies

For the majority of immunostaining experiments, E7.5 and E8.0 embryos were dissected in cold PBS and fixed for 1 hr in 4% PFA, with the exception that 8% PFA was used for detection of phosphorylated myosin. Wholemount embryos were blocked overnight in blocking buffer (PBS; 0,1%Triton; 3%BSA), incubated overnight at 4°C with primary antibodies, washed in blocking buffer, and incubated overnight at 4°C with species-specific secondary antibodies. For staining of F-actin embryos were incubated with Phalloidin-FITC/TRITC (Invitrogen).

The following antibodies were used: rabbit anti-myosin heavy chain IIB (1/400, Biolegend 909902), mouse anti-myosin heavy chain IIB (1/50, DSHB CMII 23), rabbit anti-pMRLC (1/200, Cell Signaling Ser19 3671), mouse anti-pMRLC (1/200, Cell Signaling Ser19 3675), rabbit anti-ppMRLC (1/200, Cell Signaling Thr18/Ser19 3674), rat anti-ZO-1 (1/50, DSHB R26.4C), mouse anti-ZO-1 (1/200, Invitrogen 33-9100), rabbit anti-ZO-1 (1/200, Zymed 61-7300), mouse anti-aPKC $\lambda$ (1/100, BD Bioscience 610207), chick anti-GFP (1/400, Abcam 13970). Rabbit anti-Crumbs2 was a gift from Jane McGlade (Hospital for Sick Children, Toronto, Canada) and was used at 1/50 (**Laprise et al., 2006**). Rabbit anti-PatJ was a gift from Andre Le Bivic (Developmental Biology Institute of Marseille, France) and was used at 1/200 (**Lemmers et al., 2002**). Rat anti-Rock1 was a gift from Masatoshi Takeichi (RIKEN Center for Developmental Biology, Kobe, Japan) and used at 1/100 (**Nishimura and Takeichi, 2008**). Guinea Pig anti-Celsr1 was a gift from Danelle Devenport (Princeton University, Princeton, NJ, USA) and was used at 1/500 (**Devenport and Fuchs, 2008**). Fluorescent secondary antibodies Alexa 488, 568, and 647 from Life Technologies and Thermo Fisher were used at 1/500.

### Fixed embryo epiblast mounting and imaging

Following staining, embryos were microdissected to retain the posterior side containing the primitive streak and mounted on glass slide and coverslip in Fluoromount-G, with the cavity and apical surface of the epiblast facing the coverslip. Layers of tape were placed on both extremity of the slide to create a slight volume and to avoid extensive compression and flattening of the tissue. The position of the primitive streak was determined to coincide with the posterior midline running from the allantois to the node (two morphologically distinguishable structures), and streak axis was oriented so that the proximo-distal axis was oriented along the vertical axis of the image. Most immunostaining revealed a high concentration of puncta also allowing us to identify the location of the primitive streak after imaging. Embryos were imaged on inverted Leica SP8 or Zeiss LSM880 laser point scanning confocal using 40× oil immersion objectives, with the apical surface of epiblast cells closest to the objective, and images were acquired as Z-stacks with 1 μm step.

### Segmentation and quantification of fixed embryo staining

Confocal images were analyzed using ImageJ. The primitive streak region was located and a maximal intensity projection of a minimal thickness (5–10 μm) was generated to observe the apical surface of cells in a selected region of interest at the primitive streak. ZO-1 localization was generally used as an apical junction marker for segmentation.

Segmentation was performed using the Tissue Analyzer software (**Aigouy et al., 2010**; **Aigouy et al., 2016**), an ImageJ plugin which uses a watershed algorithm to segment the cell cortex. Briefly,

a first membrane segmentation was automatically performed, and then verified and corrected manually. Cells and junctions were tracked and assigned IDs. Most cell parameters used were calculated by the software, including apical surface area, cell elongation magnitude and orientation, number of junctions per cell, and signal intensities along junctions. In brief, cell surface elongation is characterized by an axis and magnitude, and can be represented by a symmetric tensor at the centroid of a cell (images with red nematic bars). Calculation integrates cell area and shape so that cells of similar shape with different sizes are assigned the same value of cell elongation (see *Aigouy et al., 2016*, for detailed equations).

Color-coded representations of apical surface area, surface elongation, and junction number were generated in Tissue Analyzer. Surface elongation index (magnitude) calculated by the software was represented as an index from 0 (round) to 1 (line). Surface elongation and planar polarity of proteins were characterized by an axis and magnitude, represented by a symmetric tensor at the centroid of a cell (images with red nematic bars).

Intensity measurements of immunostaining are presented for every data as the mean intensity along single junctions and normalized to the global mean junctional intensity of corresponding embryos. Anisotropy parameters were calculated from junctional intensities measured within Tissue Analyzer. Intensities were measured along three pixel-wide lines (~500 nm), measurements were extracted from a database created by the software and used externally for calculations. For each single cell analyzed, the mean junction intensity was used to calculate per cell, the SD between the intensity of all junctions, and the junction with the maximal intensity divided by the junction with the minimal intensity (Max/Min ratio). These two parameters were used to evaluate the anisotropic accumulation of proteins at apical junctions. Double immunostainings were used to assess correlative accumulation of proteins, and mean junction intensities were used to create scatter plots to evaluate the dispersion of the intensity across the populations, and their correlation coefficients. Generally, measurements taken of junctions from different embryos were pooled, except for a few examples in which junctions were grouped by individual cells to evaluate the range of anisotropy in different cells.

For planar polarity of protein localization at apical junctions, Tissue Analyzer calculates a planar polarity characterized by an axis and magnitude, and can be represented by a symmetric tensor at the centroid of a cell (images with red nematic bars). Cell junctions fluorescent intensity as well as the angles of junction and the geometry of the cell were used to define the magnitude of polarity as well as the angle (see *Aigouy et al., 2016*, for detailed equations). For protein accumulation based on junctional orientation, average junctional intensities were plotted according to junctional orientation angles which were classified in bins of 15°.

For protein quantifications in *Crb2*$^{-/-}$ mutants and comparison with WT embryos, three to four embryos of each genotype recovered from two to three litters where processed, immunostained and imaged in parallel using the same parameters, then segmented and quantified as described above. Measurements are presented as the mean intensity along single junctions and normalized to the global mean junctional intensity of the pool cohort of embryos of both genotypes for each proteins. For F-actin quantification, junctional values represent the intensity measured at the peripheral cortex and the medial values the intensity measured in the medial/cytoplasmic area of the same cells. For quantifications of myosin heavy chain IIB, ppMRLC, aPKC, and Rock1, junctional signal intensities were normalized to the cytoplasmic medial signal intensities.

## Time-lapse imaging

For time-lapse imaging, mid- to late-streak stage embryos (E7.5) were dissected with their yolk sac and ectoplacental cone intact in dissection medium ($CO_2$-independent DMEM, 10%FBS) at 37°C. Embryos were stabilized in a depression created in a layer of collagen matrix and imaged in glass-bottom 35 mm MatTek dishes. They were placed with their posterior side facing the objective and culture in 50% rat serum/50% DMEM media in an incubation chamber at 37°C and 5% $CO_2$. Embryos were imaged for a total of 3–6 hr with Z-stacks of 1 μm step acquired every 5 min (minimum time-scale possible to keep embryo and fluorescent signal integrity) on a Leica SP8 laser point scanning confocal or a Nikon A1RHD25 high-speed resonant scanning confocal with 40× objectives. Imaging usually focused on regions of interest in the proximo-distal middle of the primitive steak where the epiblast is the least curved and optimal for image data acquisition and analysis. Since imaging the apical surface of epiblast cells in living mouse embryos is challenging, we evaluated time-lapse experiments

on several different microscope systems (laser point scanning confocal, spinning disc, light-sheet). The high-speed resonant scanning confocal (Nikon A1R) allowed us to obtain the best spatial and time resolution, while keeping embryos alive and preserving fluorescent signal integrity. Under these conditions, using the Nikon A1R system, we were unable to acquire images with time resolutions shorter than 5 min.

## Dynamic images analysis and quantification

Membrane-GFP from *mT/mG* embryos was used to observe apical constriction associated with cell ingression at the primitive streak. For all other imaging and quantification, a ZO-1-GFP knock-in reporter was used as it generally gave the best signal-to-noise enabling observation of apical surfaces of epiblast cells. Analyses were performed in ImageJ and Tissue Analyzer on maximal projections of Z-stacks. Cell tracking in the plane of the epithelium was performed manually using the Manual Tracking plugin in ImageJ. To quantify ingression events, small areas of interest in the vicinity of the primitive streak, corresponding to an ~40 μm region along the predicted midline, were analyzed. All cells were tracked for at least 1 hr, and constriction and ingression events were identified to calculate the ratio of ingression events per hour. Ingression events were considered isolated when the constriction and disappearance of a cell occurred within 30 min or more of the ingression of adjacent neighboring cells, and clustered when occurring within a 30 min window of the ingression of adjacent neighboring cells (two to four cells in different clustered events).

For dynamic analysis of apical surfaces, small regions of interest at the primitive streak were used for segmentation with Tissue Analyzer. A first segmentation was automatically generated by the software, then each time point manually corrected. Cell parameters, including apical surface area, surface elongation index, and orientation and number of edges, were extracted from the segmentation database, and graphs were plotted using GraphPad Prism.

The beginning of cell constriction associated with an ingression event was defined as the point when cells started to reduce their apical surface area after a period of oscillation, and so continually until they completely lost their apical domain. The start of constriction was defined as t0 when the apical surface area, rate of change of surface area, and other parameters were plotted against time. Changes in apical surface area (constriction rate) was defined as the inverse value of the derivative of apical surface area: $\Delta area(t)=area(t-1) - area(t)$. Oscillations of surface area in epiblast cells away from the primitive streak and before the initiation of ingression were compared with the more dramatic changes in surface area during the constriction period. During ingression, a constriction pulse was defined as an event in which the contraction rate exceeded one SD above the mean of the contraction rate of epiblast cells (above 0.7 μm$^2$/min). The number of pulses, occurrence of pulse, and pulse magnitude were quantified manually for each constriction period of each cell and each pulse. The calculated apical area rate of change was integrated in the Tissue Analyzer database, and used to generate color-coded movie representations of surface area, area rate of changes, cell elongation, number of junctions, and junction length. As for fixed embryo analysis, surface elongation index (magnitude) calculated by the software is represented as an index from 0 (round) to 1 (line). Surface elongation is characterized by an axis and magnitude, and is represented by a symmetric tensor at the centroid of a cell (red nematic bars).

Changes in junction length (shrinkage rate) was defined similarly to cell area changes, as the inverse value of the derivative of junction length: $\Delta length(t)=length(t-1) - length(t)$. To analyze and compare junctions length changes over time, correlation of pairs of junctions were quantified and plot as correlation matrix to illustrate a single cell or correlation coefficient plot for pool of single cells. The average shrinkage rate of single junctions during consecutive cell constriction pulses and stable phases was calculated to visualize the asynchronous and differential shrinkage of junctions during single-cell constriction.

## Statistics

Three to six embryos were analyzed for each experiment. Details of n values, means, and p values can be found in *Supplementary file 1*. Error bars on graphs represent s.e.m. p Values correspond to an unpaired non-parametric Mann-Whitney test for comparison of pairs of conditions, or one-way ANOVA for multiple comparisons. All cells were included in the statistical analysis. Each embryo was considered a biological replicate. No randomization or blind analysis was conducted.

## Definition of terms

### Start of constriction

The beginning of cell constriction associated with an ingression event at the primitive streak was defined as the time point when cells started to reduce their apical surface area after a period of oscillation, and so continually until they completely lost their apical domain.

### End of constriction

The end of constriction was defined as the time point when the constricting cells lost their apical surface and move out of the plan of the epiblast.

### Before constriction

Defines the time-period before cells start their constriction at the primitive streak. This time-period shows same oscillation characteristics of the apical surface area as epiblast cells further away from the primitive streak.

### Constriction

Defines the time-period during which cells reduced their apical surface area until they completely lose their apical domain. We also use the term for the constriction rate and constriction pulse.

### Contraction/expansion

Used to define the rate of change in apical surface area through time, contraction defining the rate of reduction in area, and expansion a rate of increase in area. They are defined both as positive value and plotted as rate of change, or in some cases defined negatively and positively and plotted as constriction rate.

### Constriction pulse

Defines the time-period during the constriction when the rate of contraction of the apical surface area exceeds the threshold defined as one SD above the mean of the contraction rate of epiblast cells (above 0.7 $\mu m^2$/min).

### Stable phases

Defines the time-period between constriction pulses, when the change in apical surface area shows minimal variation.

### Pulse occurrence

Defines the frequency at which constriction pulses are occurring.

### Pulse magnitude

Defines as the maximum value of constriction rate within each pulse.

### Shrinkage

Defines the junction reduction in length, shrinkage rate indicates the change in junction length through time.

## Acknowledgements

AF and AKH dedicate this manuscript to the memory of our late colleague and mentor Kathryn V Anderson. Kathryn marveled at the spectacle of mammalian gastrulation, and recognized the insights that genetics and imaging would bring. We thank Andre Le Bivic, Danelle Devenport, Jane McGlade, and Masatoshi Takeichi for antibodies, Terry Lechler and Robert Adelstein for mice. We thank Zhirong Bao, Alex Joyner, Jen Zallen, and members of the Hadjantonakis lab for stimulating discussions and critical feedback on the work. We thank the MSKCC (Memorial Sloan Kettering Cancer Center) Molecular Cytology, and the SKI Light Microscopy Instrument Cluster and James Muller for imaging technical

support. This study was supported by the NIH (R01HD094868, R01DK127821, R01HD086478, and P30CA008748). AF was supported by a postdoctoral fellowship from the MSKCC GMTEC (The Alan and Sandra Gerry Metastasis and Tumor Ecosystems Center) for part of this work.

## Additional information

### Funding

| Funder | Grant reference number | Author |
|---|---|---|
| National Institutes of Health | R01HD094868 | Alexandre Francou<br>Anna-Katerina Hadjantonakis<br>Kathryn V Anderson |
| The Alan and Sandra Gerry Metastasis and Tumor Ecosystems Center of MSKCC | Postdoctoral Fellowship | Alexandre Francou |
| National Institutes of Health | P30CA008748 | Alexandre Francou<br>Anna-Katerina Hadjantonakis<br>Kathryn V Anderson |
| National Institutes of Health | R01HD086478 | Alexandre Francou<br>Anna-Katerina Hadjantonakis<br>Kathryn V Anderson |
| National Institutes of Health | R01DK127821 | Alexandre Francou<br>Anna-Katerina Hadjantonakis<br>Kathryn V Anderson |

The funders had no role in study design, data collection and interpretation, or the decision to submit the work for publication.

### Author contributions
Alexandre Francou, Conceptualization, Formal analysis, Funding acquisition, Investigation, Writing - original draft, Writing - review and editing; Kathryn V Anderson, Conceptualization, Resources, Supervision, Funding acquisition; Anna-Katerina Hadjantonakis, Conceptualization, Resources, Supervision, Writing - review and editing

### Author ORCIDs
Alexandre Francou ⓘ http://orcid.org/0000-0002-6295-3777
Anna-Katerina Hadjantonakis ⓘ http://orcid.org/0000-0002-7580-5124

### Ethics
All animal experiments in this study were conducted in accordance with with the Guide for the Care and Use of Laboratory Animals of the National Institutes of Health and within the guidelines of the Memorial Sloan Kettering Cancer Center (MSKCC) Institutional Animal Care and Use Committee (IACUC) under protocol number 03- 12-017.

### Decision letter and Author response
Decision letter https://doi.org/10.7554/eLife.84019.sa1
Author response https://doi.org/10.7554/eLife.84019.sa2

## Additional files

### Supplementary files
- MDAR checklist
- Supplementary file 1. Details of the n values and p values presented in the figures.

## Data availability

All data generated and quantified in this study are included in the Source data files.

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
