## [Editor Report]

This study employs live imaging to investigate the movement of mesodermal cells in early mouse embryos. By examining the dynamics of cell behavior in normal and mutant embryos, the authors propose that apical constriction of cells results from pulsed contraction guided by crumbs2 signals. The paper presents beautiful images and adds to the molecular understanding of cell migration during early development.

---

## [Decision Letter]

**Decision letter after peer review:**

Thank you for submitting your article "A ratchet-like apical constriction drives cell ingression during the mouse gastrulation EMT" for consideration by *eLife*. Your article has been reviewed by 3 peer reviewers, and the evaluation has been overseen by a Reviewing Editor and Marianne Bronner as the Senior Editor. The following individuals involved in the review of your submission have agreed to reveal their identity: Ann Sutherland (Reviewer #1); Jean-Léon Maître (Reviewer #2).

Essential revisions:

This is a nicely done paper with high-quality movies and images, representing a major strength. The main weakness is that many of the claims are based on somewhat unsubstantiated correlations that require further validation. We refer you to the full reports of the reviewers for details.

1. The issues of temporal resolution and IF quantitation as outlined in the reviews are important to address.

2. The crumbs2 -/- embryos should be more carefully looked into. This would require extra analyses.

3. The discrepancies between the findings here and previous publications on crumbs2 should be addressed as they lead to contradictory interpretations.

*Reviewer #1 (Recommendations for the authors):*

The authors have done an exceptional job of documenting both the cell behavior and the patterns of protein localization and their correlation. It is a clear model for the rest of the field to follow in evaluating these aspects of development in any system.

This meticulous approach is not as clearly evident in the data on the Crumbs2 mutant embryo. Here the authors show localization data, but no live imaging to be able to compare the behavior in the mutant to the wild type. It would seem obvious, given the data obtained from the WT, that a comparable evaluation of the Crumbs2 mutant might show which aspect of the behavior is affected, and thus where Crumbs2 is having its effect. In addition, the findings of the previous publication on Crumbs2 (Ramkumar et al., 2016, also from the Anderson lab) do not support the model proposed here, and in fact, contradict it. The authors should discuss this at the very least and might want to alter their model for the action of Crumbs2 based on it. Related to this, the statement on p18, line 461 that "Epiblast cells fail to ingress…" in the mutant should be more nuanced – they do ingress early, but appear to fail at later stages. Data on the difference in ingression behavior between these stages would be really valuable. This may be beyond the scope of this manuscript, but should be considered. There are many aspects to the Crumbs2 mutant that are not well explained, and seem to be glossed over in favor of a simple explanation.

*Reviewer #2 (Recommendations for the authors):*

The imaging performed by the authors is fantastic and the analysis is very insightful. However, with a 5min temporal resolution of a process that takes 25-90min, there is only so much information that can be extracted. Pulsed contractions in fly embryos last between 50-120 s. Without a sufficient temporal resolution, they could not be detected. Here, with a 5min resolution, it becomes difficult to distinguish pulses from measurement noise. Have the authors measured the accuracy of their segmentation? Could segmentation errors be sufficiently large to be mistaken for a constriction pulse or a stabilization? How much of a surface area change do they consider meaningful to distinguish between constriction and stable phases? Have the authors run shorter time lapses with higher temporal resolution (30s to 1min interval max) to verify their claim? This potential limitation becomes most prominent in F3 when analysing individual junctions from a single cell. How can the authors distinguish between asynchronous shrinkage from the data shown in the figure and insufficient temporal resolution coupled with segmentation errors? Also in this figure, it seems that only one cell from one embryo is considered to draw a general conclusion. Have the authors analysed more cells and embryos?

Immunostaining is not a quantitative method, even if the staining and imaging are performed in the same conditions. When comparing intensity between WT and Crumbs2 mutants, how do the authors know that they are not measuring a difference in light penetration through the tissue (for example if the mutants would be thicker than WT)? Fluorescence intensities need to be normalized within objects within the same confocal plane (provided that illumination is homogeneous). It seems that the junction to the medial signal is affected by actin F7B. Why not use the same kind of normalization for other markers? If the authors want to claim that there would be less myosin, aPKC, Rock1 in Crumbs2 mutants, they need to take a different approach (ex: qPCR, Western blot). In the end, I think the authors make their point with F7D. Are the SD and max/min values normalized to the mean value of the batch of embryos as well? If mGFP data is not available for Crumbs2 mutant, this data should be removed or made more explicitly clear that it only refers to WT embryos.

Regarding the colocalization analysis, the authors detect little to no correlation between myosin and crumbs. They then refer to these proteins as being "complementary". Yet in their schematic in F5E, they represent them as mutually exclusive. I think this schematic and the term "complementary" are exaggerated. Complementarity or mutual exclusion, which are synonyms, would require an anti-correlation between the signals. The authors argue that the noise in the data supports a dynamic localization of the proteins. Currently, I do not think this is supported by the data. Are ZO1-GFP levels dynamic? Could the authors use their ZO1-GFP movies and immunostaing of ZO1 to measure the dynamics of this protein between live and immunostaining?

*Reviewer #3 (Recommendations for the authors):*

The data reported here are quite valuable to researchers interested in mammalian gastrulation (although they are largely descriptive). To substantiate the authors' observations, analysis-wise, I would like to see some perturbation data and time-lapse data with at least one other junctional marker. Could the authors also comment on spatio-temporal variability (my understanding is that what they looked at was restricted to the mid-streak level and to a 3-6 hour window between E7.5-E8)? Do authors imply that this is a generalizable mode of epiblast ingression in the mouse embryo? If so, how can dynamic apical constriction be coordinated with other junctional and cell-matrix remodeling necessary for ingression? Overall, I think the authors presented high-quality imaging data, and that the observations reported in this manuscript (in its current form) are intriguing (if somewhat descriptive).

[Editors' note: further revisions were suggested prior to acceptance, as described below.]

Thank you for resubmitting your work entitled "A ratchet-like apical constriction drives cell ingression during the mouse gastrulation EMT" for further consideration by *eLife*. Your revised article has been evaluated by Marianne Bronner (Senior Editor) and a Reviewing Editor.

The manuscript has been improved but there are some minor remaining issues that need to be addressed, as outlined in the reviews below:

*Reviewer #1 (Recommendations for the authors):*

I find the revised manuscript to be excellent, and I feel the authors responded well to the prior reviews. I would have liked to see live imaging of the Crumbs2 mutant, as I think the authors would learn a lot about the nature of the change in morphology of these cells, and about the process of cells coming to the streak more generally. One thought is that the ingression is what passively recruits new cells to the streak, but the crowding documented by Francou et al. suggests other, more active mechanisms. I do understand that overcoming logistical impediments to imaging these embryos might not be possible, and I think the data presented from immunostaining provides a great deal of understanding of the effects of the mutation.

One issue that the authors should check is in Figure 7D – there are 5 sets of bars but 6 labels below, and it is difficult to determine whether MyoIIB or GFP is represented in the final two bars.

Otherwise, I find the manuscript to be acceptable for publication.

*Reviewer #2 (Recommendations for the authors):*

I thank the authors for addressing most of the points of the reviewers and for modifying their manuscript accordingly.

Regarding immunostaining analyses, I agree with the authors that there is no easy alternative method to compare quantitatively the subcellular localization of proteins in WT and Crumb2 mutants. Nevertheless, the absence of alternatives does not warrant the use of semi-quantitative analyses for quantitative claims. My point was not to suggest doing another type of quantitative experiment but to adapt the conclusions based on the type of method used. With these analyses, the authors can claim, for example, that the junction/medial ratio of actin is reduced in Crumb2 mutant compared to WT (based on the data shown in F7B, this should be correct). However, they cannot claim "reduced levels of Myosin Heavy Chain IIB and ppMLC (the fully activated form of Myosin Light Chain) in mutants at E7.5 (Figure 7A, B)" with a semi-quantitative method.

I would advise normalizing the junctional signal to the medial one (to correct for depth-related artefacts, which is very likely to play in this kind of sample) and to show this data for WT and mutants. This will allow for a comparison of relative levels, which is what matters for apical constriction in the end.

"Myosin light chain" (MLC) is ambiguous. The authors presumably refer to the regulatory light chain and not the essential light chain. The author could use MRLC as an acronym as most often used in the literature.

---

## [Author Response]

Essential revisions:This is a nicely done paper with high-quality movies and images, representing a major strength. The main weakness is that many of the claims are based on somewhat unsubstantiated correlations that require further validation. We refer you to the full reports of the reviewers for details.1. The issues of temporal resolution and IF quantitation as outlined in the reviews are important to address.2. The crumbs2 -/- embryos should be more carefully looked into. This would require extra analyses.3. The discrepancies between the findings here and previous publications on crumbs2 should be addressed as they lead to contradictory interpretations.

We thank the handling editor Dr Bronner and all three reviewers, especially Drs Maitre and Sutherland, for their careful evaluation of our work and for their constructive critiques towards its improvement.

We have endeavored to address all the reviewers’ comments, with our revisions having included substantial re-analyses as well as additional experiments.

We have addressed all three major points listed above in our revised manuscript. In particular, we have undertaken a detailed reanalysis of our data, so that any conclusions made are better supported by the data analyses.

In places we were unable to address specific points, for example where there were technical limitations precluding the acquisition of new data, we provide detailed responses to justify omissions, and detailed discussions relating to our continued efforts.

Reviewer #1 (Recommendations for the authors):The authors have done an exceptional job of documenting both the cell behavior and the patterns of protein localization and their correlation. It is a clear model for the rest of the field to follow in evaluating these aspects of development in any system.This meticulous approach is not as clearly evident in the data on the Crumbs2 mutant embryo. Here the authors show localization data, but no live imaging to be able to compare the behavior in the mutant to the wild type. It would seem obvious, given the data obtained from the WT, that a comparable evaluation of the Crumbs2 mutant might show which aspect of the behavior is affected, and thus where Crumbs2 is having its effect.

We appreciate the reviewer’s comment and agree that live imaging of *Crumbs2* mutants will provide insight on the role of Crumbs2. However, due to the technically challenging nature of the live imaging experiments performed for this study, for example the apical surfaces of epiblast cells being at the farthest point from the imaging objective, and the dim signal of the fluorescent reporters being used, we were unable to acquire data on the necessary n=3 stage-matched *Crumbs2* mutant embryos.

Recognizing this gap in knowledge and to make such experiments feasible, we are establishing an alternative image acquisition pipeline making improvements in image data acquisition in particular using alternative microscopic imaging modalities (for example using light-sheet microcopy). However, these methods are still being optimized for our samples. Moreover, we believe their application is beyond the remit of this study, and the timeframe for publication.

In addition, the findings of the previous publication on Crumbs2 (Ramkumar et al., 2016, also from the Anderson lab) do not support the model proposed here, and in fact, contradict it. The authors should discuss this at the very least and might want to alter their model for the action of Crumbs2 based on it.

We thank the reviewer for bringing up this aspect as it is an important point that we now discuss in greater detail and clarify in the revised manuscript.

Importantly, we do not believe our data are in disagreement with the previous study of Ramkumar et al. The precise details of the defect observed in *Crumbs2* mutants are still not totally clear. However, we would like to point out that in Ramkumar et al., the timelapse data did not show cells constricting their surfaces, but rather these data revealed that cells having small apical surfaces failed to detach and ingress out of the epiblast layer. Thus, this previous study focused on the subsequent step in the process to that being addressed in the present work.

Furthermore, epiblast cells outside the domain occupied by the primitive streak and even some cells positioned on the lateral sides of the embryo were reported to exhibit abnormally small apical surfaces. These cells will not normally constrict their apical surfaces, since they are not going to undergo the gastrulation EMT, which is a behavior restricted to the region of the primitive streak. In sum, these previous data do not directly address nor demonstrate that epiblast cells in *Crumbs2* mutants undergo apical constriction.

Moreover, in *Crumbs2* mutants a large number of cells were reported to fail to ingress at the primitive streak, and consequently they were seen to accumulated within the epiblast epithelial layer. Indeed, we believe that the small apical surfaces observed in *Crumbs2* mutants, are most likely to result from the crowding/jamming of cells within the epiblast layer, and that this causes changes in the shape and volume of cells due to them being spatially constrained. Thus, increased crowding of epithelial cells within a spatially constrained tissue, likely drives reduced apical surfaces and extensive apico-basal elongation, as observed in *Crumbs2* mutants.

Related to this, the statement on p18, line 461 that "Epiblast cells fail to ingress…" in the mutant should be more nuanced – they do ingress early, but appear to fail at later stages.

The reviewer makes a good point regarding the *Crumbs2* mutant phenotype. It is true that in this mutant, a first wave of gastrulation EMT is observed at E6.5. We interpret this to mean that Crumbs2 is not required at the onset of the gastrulation EMT, as the first cells ingress through the primitive streak. Consequently, a small number of mesodermal cells are produced. However, ingression defects in *Crumbs2* mutant embryos are evident within 24hours and are evident by E7.75.

These distinct sequential phases of gastrulation regulation, initially independent of Crumbs2, but subsequently dependent, were not initially discussed in our manuscript for simplicity, but we have now elaborated these details in the revised manuscript.

Data on the difference in ingression behavior between these stages would be really valuable. This may be beyond the scope of this manuscript, but should be considered.

We feel that these experiments would be exceptionally valuable, and would constitute a subsequent study.

In the present study, giving the technical difficulties of the live imaging, we focused our data analysis on the E7.5 stage of development when the primitive streak is fully elongated to capture as many ingression events as possible. This is also the stage when the *Crumbs2* mutants exhibit defects.

However, we recognize that it will be interesting and indeed comprehensive for us to analyze ingression behavior at stages prior to and after this time-point, but due to the difficulty of the imaging methods applied in this study, we chose to focus on one developmental stage. Such extended analyses are beyond the scope of the current study, and will only be performed once we have alternative imaging methods established and protocols are optimized for these serial gastrulation stages.

There are many aspects to the Crumbs2 mutant that are not well explained, and seem to be glossed over in favor of a simple explanation.

We acknowledge that the phenotype of the *Crumbs2* mutants is complex, and we have elaborated our discussion of several aspects in the revised manuscript.

Reviewer #2 (Recommendations for the authors):1. Some inaccuracies and confusing termsAbstract:– When epiblasts cells ingress to become mesoderm, I think it is confusing to state that they would "lose pluripotency". Mesodermal cell differentiation potential may be reduced but they clearly remain pluripotent.

The statement has been edited in the revised manuscript.

– "stochastic-like" is very confusing. What do the authors mean?

We agree regards the confusion of the term used, and this has been edited.

– "complementary" may be confusing as well, see below.

We agree and now use the term “reciprocal enrichment” in the revised manuscript to

refer to the accumulation of proteins on different junctions.

Introduction:– EMT does not occur in all triploblasts. For example, zebrafish embryos do not separate their germ layers via EMT since the deep cells are not epithelial in the first place.

This is correct and has been edited in the revised manuscript.

– In Samarage et al., they describe the constriction of cells in preimplantation embryos to form the ICM but this is not an apical constriction since the cells that constrict are those without apical membranes.

We thank the reviewer for bringing this up. This has been edited in the manuscript.

– "Myosin-2" and "Myosin-2B" can be confusing. These are Non-muscle myosin II and Non-muscle myosin heavy chain IIB (or Myh10, as for the latest nomenclature).

This has been edited throughout the revised manuscript. Myosin Heavy Chain IIB was used when referring to the latest nomenclature (Myh10). In a few cases, Myosin II was used when referring more generally to Non-muscle Myosin II, in *Drosophila*, or when grouping Myosin Heavy Chain IIA and IIB, or when grouping Myosin Heavy Chain IIB and phosphorylated-MLC.

– The statement about fly and mouse mesoderm ingression not being comparable because of different time-scale is not very convincing and the numbers brought by the authors throughout the manuscript are confusing. Sometimes, the authors claim that it takes 2 days, 1 day, and 25min. It is confusing to mix the entire gastrulation process and the apical constriction. Apical constriction in fly takes 10min and 25-90min in mouse. This is very much comparable.

We acknowledge that the different time scales we refer to pertaining gastrulation and apical constriction, were confusing as discussed. We have modified the sentences in which we discuss these to make it less confusing. The duration of the apical constriction we observed in the mouse epiblast can be up to 9 times greater compared to in *Drosophila*. Thus, in our opinion it is difficult to conclude whether such time scales are comparable or not.

Methods:– The "definition of terms" section is very useful. It would benefit from providing objective descriptors. For example, what value must the constriction rate reach to be part of a pulse, vs a stable phase?

This was presented elsewhere in the methods, and we have now included it in the definition of terms.

2. The imaging performed by the authors is fantastic and the analysis is very insightful. However, with a 5min temporal resolution of a process that takes 25-90min, there is only so much information that can be extracted. Pulsed contractions in fly embryos last between 50-120 s. Without a sufficient temporal resolution, they could not be detected. Here, with a 5min resolution, it becomes difficult to distinguish pulses from measurement noise. Have the authors measured the accuracy of their segmentation? Could segmentation errors be sufficiently large to be mistaken for a constriction pulse or a stabilization? How much of a surface area change do they consider meaningful to distinguish between constriction and stable phases? Have the authors run shorter time lapses with higher temporal resolution (30s to 1min interval max) to verify their claim? This potential limitation becomes most prominent in F3 when analysing individual junctions from a single cell. How can the authors distinguish between asynchronous shrinkage from the data shown in the figure and insufficient temporal resolution coupled with segmentation errors? Also in this figure, it seems that only one cell from one embryo is considered to draw a general conclusion. Have the authors analysed more cells and embryos?

We acknowledge the reviewer’s comments and agree that a shorter time resolution would be ideal to detect constriction pulses of apical surfaces. However, we need to take into account that imaging the apical surface of cells within the epiblast layer, which constitutes the most internal surface inside the embryo, is a very challenging task to perform in a gastrulating mouse embryo.

As suggested by the reviewer, we attempted to image with a shorter time interval than 5min on several different microscope systems and modalities available at our institution (including laser point scanning confocal, spinning disc, light-sheet microscopes) and were not successful in acquiring usable images with the ZO1-GFP knock-in reporter. We also need to take into account that these single-copy GFP knock-in reporters are often dim, thereby exacerbating the issue. In our hands, a high-speed resonant scanning confocal (Nikon A1RHD25) is the system that gave us the best signal-to-noise ratio, spatial resolution and temporal resolution, and was the one used for our most recent movies. Using this system, we were able to acquire a limited number of time-lapses with a time resolution of 2min, but none with a shorter time interval, and from our analyses 2min time intervals did not yield increased detail over 5min time intervals to warrant a detailed analysis.

We have provided a detailed discussion of these issues within the revised manuscript and edited some of the conclusions.

A constriction pulse was defined as an event in which the contraction rate exceeded one standard deviation above the mean of the contraction rate of epiblast cells (above 0.7 μm2/min).

The asynchronous reduction of junctions has been analyzed in several cells from different embryos. Figure 3 shows an example of one cell, we have now added another example in Figure 3 —figure supplement 1. Graph F in this figure shows the correlation of reduction of junctions in multiple cells from different embryos.

3. Immunostaining is not a quantitative method, even if the staining and imaging are performed in the same conditions. When comparing intensity between WT and Crumbs2 mutants, how do the authors know that they are not measuring a difference in light penetration through the tissue (for example if the mutants would be thicker than WT)? Fluorescence intensities need to be normalized within objects within the same confocal plane (provided that illumination is homogeneous). It seems that the junction to the medial signal is affected by actin F7B. Why not use the same kind of normalization for other markers? If the authors want to claim that there would be less myosin, aPKC, Rock1 in Crumbs2 mutants, they need to take a different approach (ex: qPCR, Western blot). In the end, I think the authors make their point with F7D. Are the SD and max/min values normalized to the mean value of the batch of embryos as well? If mGFP data is not available for Crumbs2 mutant, this data should be removed or made more explicitly clear that it only refers to WT embryos.

We acknowledge that immunostaining is not the most quantitative method, but we were unable to come up with alternative methods that can be used in our case. We believe he junctional reduction of Myosin, aPKC and Rock1 are generally due to a nonrecruitment or activation at the junctions, and not necessarily reflecting reduced expression at the gene or protein level. We do not believe that methods such as RTqPCR or Western blotting would be informative in the context in which we are looking, especially since they do not yield spatial resolution. Furthermore, we would need to isolate primitive streak cells to consider applying these methods, and we do not believe they would provide sufficient resolution.

By contrast to the live imaging (Figure 1) which was performed by placing the objective at the posterior side of the embryo in closest proximity to the outer visceral endoderm layer, for fixed tissue imaging embryos were microdissected to recover the posterior side containing the primitive streak, and were imaged on the side of the cavity by placing the objective in closest proximity to the inner epiblast layer, for direct access to the apical surface of epiblast cells at the streak. In this fixed tissue imaging configuration, the apical surfaces of cells in WT and *Crumbs2* mutants were in closest proximity to the imaging objective and thus directly accessible, and the difference in tissue thickness on the other side of the epithelium did not interfere with light penetration. We have edited the figures to clarify how the objective positions are flipped with respect to the primitive streak regions at the embryo’s posterior for live vs. fixed tissue imaging. We have now measured the signal intensity in the cytoplasmic region of WT and *Crumbs2* mutant embryos, and junctional intensity measurements have been normalized to cytoplasmic intensities.

For the SD and Max/Min data, the intensity values have been normalized to the mean before calculating the SD and Max/Min, but these are not normalized again after calculation. The mGFP data of the SD and Max/Min has been removed from Figure 7D.

Regarding the colocalization analysis, the authors detect little to no correlation between myosin and crumbs. They then refer to these proteins as being "complementary". Yet in their schematic in F5E, they represent them as mutually exclusive. I think this schematic and the term "complementary" are exaggerated. Complementarity or mutual exclusion, which are synonyms, would require an anti-correlation between the signals.

These anisotropic accumulations of proteins are complex and not easy to explain and illustrate. We use the term “reciprocal enrichment” instead of “complementary” in the revised manuscript.

The plot of Myosin and Crumbs junctional intensities in Figure 5B show individual junctions from several embryos pooled together, and it reveals a spread distribution and no correlation. Greater precision of these complex data are shown in panel (D) in the Figure 5 which depicts single junctions grouped my cell. Figure 5D shows a mix of cells with some having a correlated accumulation, some with very little correlation and others with anti-correlated accumulation. This plot illustrates that patterns can be different among different cells, and that the accumulation of certain proteins correlates in some cells, but anti-correlates in others. We are discussing this issue in greater detail in the revised manuscript.

Figure 5E does not show mutual exclusion as some junctions accumulate both groups of proteins. To clarify our observations, we have numbered the junctions in the revised figure. Junction 1 predominantly accumulates the Myosin group with very little accumulation of the Crumbs group, junction 2 on the other hand predominantly accumulates the Crumbs group with very little accumulation of the Myosin group, while junction 3 shows an intermediate accumulation of both groups of proteins.

The authors argue that the noise in the data supports a dynamic localization of the proteins. Currently, I do not think this is supported by the data. Are ZO1-GFP levels dynamic? Could the authors use their ZO1-GFP movies and immunostaing of ZO1 to measure the dynamics of this protein between live and immunostaining?

We acknowledge the reviewer’s comment and agree that we did not directly prove that the protein patterns are dynamic. This is a hypothesis, and we have edited this point in the revised manuscript so as not to make such a strong conclusion.

We evaluated the potential to measure the intensity of the ZO1-GFP reporter in our timelapse data to uncover any dynamic changes in the levels of the protein, but concluded that the resolution of our live imaging experiments was not sufficient to perform such a quantification.

Reviewer #3 (Recommendations for the authors):The data reported here are quite valuable to researchers interested in mammalian gastrulation (although they are largely descriptive). To substantiate the authors' observations, analysis-wise, I would like to see some perturbation data and time-lapse data with at least one other junctional marker.

We performed segmentation of apical surfaces in a movie of a Myosin2B-GFP reporter that is now depicted in Figure 2 Supplement 1 panels A and B of the revised manuscript. This demonstrates apical constriction with an additional marker localized at apical junctions.

Could the authors also comment on spatio-temporal variability (my understanding is that what they looked at was restricted to the mid-streak level and to a 3-6 hour window between E7.5-E8)? Do authors imply that this is a generalizable mode of epiblast ingression in the mouse embryo? If so, how can dynamic apical constriction be coordinated with other junctional and cell-matrix remodeling necessary for ingression? Overall, I think the authors presented high-quality imaging data, and that the observations reported in this manuscript (in its current form) are intriguing (if somewhat descriptive).

This is definitely a very interesting aspect pointed raised by the reviewer. We focused our data analysis on a middle region in the proximo-distal axis of the embryo, because this is the most optically accessible and flattest region of the posterior of the embryo to analyze. We also focused on the E7.5 stage of development when the primitive streak is fully elongated so as to capture as many ingression events within a time-lapse experiment as possible. Due to the difficulties associated with live imaging the apical epiblast layer of embryos at these stages, we choose to focus our analysis to a defined region of the embryo and a defined period of time. We acknowledge that it will be interesting to analyze different regions of the primitive streak and at different stages of gastrulation in order for us to uncover a general versus more distinct modes of epiblast cell ingression, but given the technical difficulties already discussed we believe that any extended analysis is beyond the scope of the current study.

We agree that analyzing how apical constriction is coordinated with other dynamic aspects of ingression and EMT is very interesting, and could be the subject of future research.

[Editors' note: further revisions were suggested prior to acceptance, as described below.]

Reviewer #1 (Recommendations for the authors):I find the revised manuscript to be excellent, and I feel the authors responded well to the prior reviews. I would have liked to see live imaging of the Crumbs2 mutant, as I think the authors would learn a lot about the nature of the change in morphology of these cells, and about the process of cells coming to the streak more generally. One thought is that the ingression is what passively recruits new cells to the streak, but the crowding documented by Francou et al. suggests other, more active mechanisms. I do understand that overcoming logistical impediments to imaging these embryos might not be possible, and I think the data presented from immunostaining provides a great deal of understanding of the effects of the mutation.One issue that the authors should check is in Figure 7D – there are 5 sets of bars but 6 labels below, and it is difficult to determine whether MyoIIB or GFP is represented in the final two bars.Otherwise, I find the manuscript to be acceptable for publication.

We thank the reviewer for picking up the issue relating to Figure 7D. Following a suggestion by one of the reviewers, this graph was revised with the bars showing the values measured on the membrane-GFP reporter having been removed. However, we overlooked removal of the corresponding label. This has now corrected, in the latest version of Figure 7 now uploaded.

Reviewer #2 (Recommendations for the authors):I thank the authors for addressing most of the points of the reviewers and for modifying their manuscript accordingly.Regarding immunostaining analyses, I agree with the authors that there is no easy alternative method to compare quantitatively the subcellular localization of proteins in WT and Crumb2 mutants. Nevertheless, the absence of alternatives does not warrant the use of semi-quantitative analyses for quantitative claims. My point was not to suggest doing another type of quantitative experiment but to adapt the conclusions based on the type of method used. With these analyses, the authors can claim, for example, that the junction/medial ratio of actin is reduced in Crumb2 mutant compared to WT (based on the data shown in F7B, this should be correct). However, they cannot claim "reduced levels of Myosin Heavy Chain IIB and ppMLC (the fully activated form of Myosin Light Chain) in mutants at E7.5 (Figure 7A, B)" with a semi-quantitative method.I would advise normalizing the junctional signal to the medial one (to correct for depth-related artefacts, which is very likely to play in this kind of sample) and to show this data for WT and mutants. This will allow for a comparison of relative levels, which is what matters for apical constriction in the end.

We agree with the reviewer on this point. In fact, we made this change in the previous revision, such that junctional intensities of MHC-IIB, ppMRLC, aPKC and Rock1 in WT and Crb mutants were normalized to the medial signals. This change was noted in our responses to reviewers’ comments. We explained the change that in the methods section, but perhaps it was not clear enough. We have now edited the text, figures and legends to further clarify the point.

"Myosin light chain" (MLC) is ambiguous. The authors presumably refer to the regulatory light chain and not the essential light chain. The author could use MRLC as an acronym as most often used in the literature.

We agree with this comment from the reviewer, and have accordingly edited our revised manuscript.